# FLOW EQUIVARIANT WORLD MODELING FOR PARTIALLY OBSERVED DYNAMIC ENVIRONMENTS

## ABSTRACT

Embodied systems experience the world as 'a symphony of flows': a combination of many continuous streams of sensory input coupled to self-motion, interwoven with the motion of external objects. These streams obey smooth, time-parameterized symmetries, which combine through a precisely structured algebra; yet most neural network world models ignore this structure and instead repeatedly re-learn the same transformations from data. In this work, we introduce 'Flow Equivariant World Models', a framework in which both self-motion and external object motion are unified as one-parameter Lie group 'flows'. We leverage this unification to implement group equivariance with respect to these transformations, thereby sharing model weights over locations and motions, eliminating redundant re-learning, and providing a stable latent world representation over hundreds of timesteps. On both 2D and 3D partially observed world modeling benchmarks, we demonstrate Flow Equivariant World Models significantly outperform comparable state-of-the-art diffusion-based and memory-augmented world-modeling architectures, training faster and reaching lower error – particularly when there are predictable world dynamics outside the agent's current field of view. We show that flow equivariance is particularly beneficial for long rollouts, generalizing far beyond the training horizon. By structuring world model representations with respect to internal and external motion, flow equivariance charts a scalable route to data-efficient, symmetry-guided, embodied intelligence. Project page: Link

## 1 INTRODUCTION

As embodied agents in a dynamic world, our survival critically depends on our ability to accurately model our surrounding environment, our own self-motion through it, and the dynamics of moving bodies within it. A natural example is pack hunting: to coordinate an attack, an agent must accurately estimate the location and velocity of a target while simultaneously predicting the motion of other pack animals. However, these world states are not simply provided to the agent in the form of an omniscient global view; instead, the agent is provided with a restricted first-person field of view that simultaneously shifts and rotates with the agent's own self-motion. The result is a highly entangled stream of stimulus flows that yields information on a fraction of the full environment at any point in time. Despite this, biological agents appear to navigate such partially observed dynamic environments effortlessly, as if they have a latent map of the environment perfectly flowing in unison with the global world state.

In this work, we study this task of *partially observed dynamical world modeling* (visualized in Fig. 1), combined with the inherent self-motion of embodied agents, and investigate if we might be able to account for both external and internal sources of visual variation in a geometrically structured manner. Specifically, we find that both internal and external motion can be understood as mathematical 'flows', enabling both sources of variation to be handled exactly as time-parameterized symmetries through the framework of 'flow equivariance' (Keller, 2025). We demonstrate that we can construct Flow Equivariant World Models that handle self-generated motion in a precisely structured manner, while simultaneously capturing the motion of external objects, even if they are moving outside the agent's field of view. We show that this yields substantially improved world modeling performance and generalization to significantly longer sequences than those seen during training, highlighting the benefits of precise spatial and dynamical structure in world models.

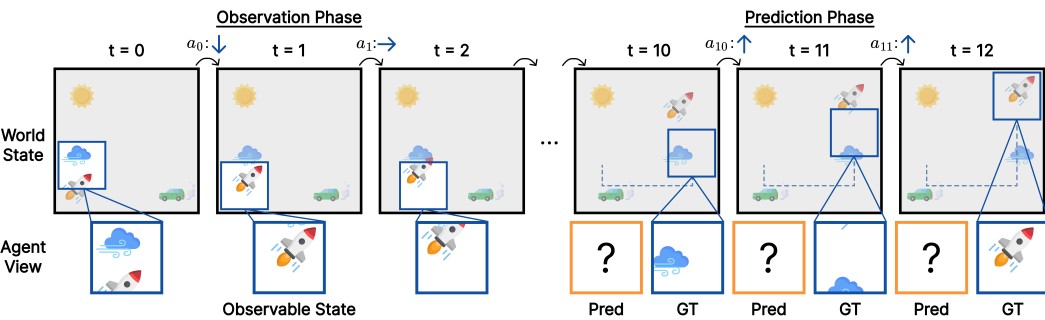

Figure 1: **Partially observable dynamic world modeling.** Grayed out areas are not visible to the agent at time $t$. The agent moves its view each timestep via action $a_t$ and must predict future states after an observation phase, conditioned on its own future action sequence.

## 2 BACKGROUND

To build world models with structured representations of environment dynamics, we rely on recent work in both world modeling and equivariance, which we briefly overview here. More thorough information on related work is available in Section 5 and Appendix F.

**World Modeling.** A world model can be described at a high level as a system affording the ability to predict not only the future state of an environment given initial conditions, but also how that state may evolve differently when acted upon by an agent (Ha & Schmidhuber, 2018). Recent work on world models has focused on representing and predicting the future world state as video, primarily using large-scale latent diffusion transformer models. While these models achieve impressive perceptual quality and scale well with growing data and compute, we argue here that their current form by design lacks the ability to predict long-horizon dynamics in partially observable environments, thus fundamentally limiting their ability to be used for real-world downstream tasks.

Partial observability is defined as a setting when the agent's observation does not contain the full information of the world's state. This problem is particularly relevant to world modeling: to accurately make predictions of the future, the model must retrieve all relevant information from previous observations, no matter how long ago it was observed, and bring it to the future prediction. Modern transformers can extend the self-attention window over many past observation frames, and recent strategies such as history guided diffusion forcing help video diffusion models utilize this context to retain self consistency (Song et al., 2025). But inevitably, as the number of observation frames grows, information must be discarded through sliding-window attention or some other approximation. This problem is exacerbated by the cost of spatiotemporal attention over a highly redundant signal such as video. Once the past observation has left the context window (true partial observability with respect to the self attention window), it has been lost; turning around will reveal an entirely new scene. Furthermore, relying on information from stale observation frames can be detrimental to modeling the natural dynamics of the evolving world around us.

Recent work has explored augmenting video diffusion models with different forms of latent memory that persist across time; however, the focus has primarily been on consistency in static 3D scenes, without a unified framework for modeling partially observed dynamics. In contrast, we argue that a natural way to build world models is with a recurrent flow equivariant memory at the core, evolving and shifting to represent both the dynamics of the world and the actions of the agent seamlessly. Such a memory enables an ability to predict future states in a precise motion-symmetric manner while maintaining important information for extended observation windows. A visual comparison between modern sliding-window transformers, existing memory solutions, and our model (FloWM) is available in Figure 2.

**Equivariance.** A neural network $\phi$ is said to be equivariant if its output, $\phi(f)$, changes in a structured, predictable manner when the input $f$ is transformed by an element $g$ of the group $G$, i.e. $\phi(g \cdot f) = g \cdot \phi(f) \ \forall g \in G$. For given choices of the linear representation of the group in the input and output spaces, $T_g$ and $T'_g$ respectively, this equation can be made more precise as: $\phi(T_g[f]) = T'_g[\phi(f)] \ \forall g \in G$. Importantly, these representations are group homomorphisms, meaning that they respect the structure of the group that they are representing; i.e. if two group

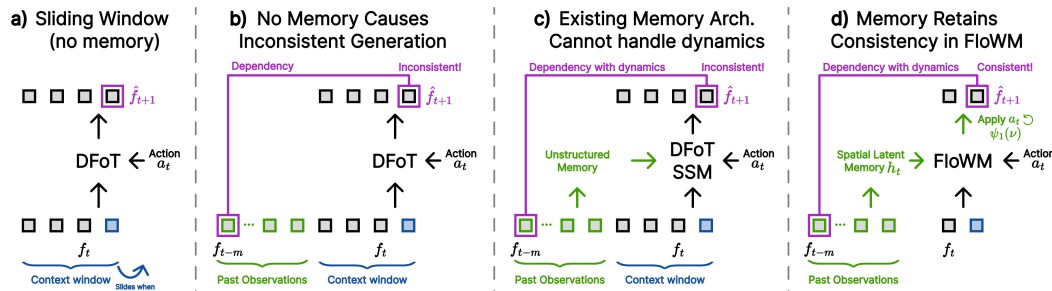

Figure 2: **Comparing World Modeling Frameworks.** a) Standard autoregressive video diffusion evicts frames beyond the sliding window. b) Information dependencies between past observations and generated frames cause inconsistency without memory. c) Existing memory solutions are view-dependent, and thus cannot predict dynamic scenes consistently. d) FloWM remembers past observations in the spatial latent memory, and continually updates them via internal dynamics.

elements combine as $g_1 \cdot g_2 = g_3$, then $T_{g_1} T_{g_2} = T_{g_3}$. This structure ensures that the equivariance condition holds for all group elements $g \in G$. One way to construct equivariant neural networks is through structured weight sharing (Cohen & Welling, 2016; Ravanbakhsh et al., 2017). This structure reduces the number of parameters that need to be learned in an artificial neural network while simultaneously improving performance by baking in known symmetries from the data distribution. For example, in the setting of molecular dynamics simulation, introducing equivariance with respect to 3-dimensional translations, rotations, and reflections (the group $E(3)$, a known symmetry of the laws of physics) increases data efficiency by up to three orders of magnitude (Batzner et al., 2022).

## 3  FLOW EQUIVARIANT WORLD MODELS

In this section, we begin with a review of Flow Equivariance, followed by an introduction of a generalized form of the recurrence relation capable of supporting complex tasks. Then, we present instantiations of our general framework for 2D and 3D partially observed dynamic world modeling.

### 3.1  GENERALIZED FLOW EQUIVARIANCE

**Flow Equivariance.**  Recently, Keller (2025) introduced the concept of flow equivariance, extending existing 'static' group equivariance to time-parameterized sequence transformations ('flows'), such as visual motion. These flows are generated by vector fields $\nu$, and written as $\psi_t(\nu) \in G$. The flow $\psi_t(\nu)$ maps from some initial group element $g_0$ to a new element $g_t$ (i.e. $\psi_t(\nu) \cdot g_0 = g_t$), and we can therefore informally think of $\psi_t(\nu)$ as a time-parameterized group element when $g_0$ is fixed. Formally, a flow $\psi_t(\nu) : \mathbb{R} \times \mathfrak{g} \to G$ is a subgroup of a Lie group $G$, generated by a corresponding Lie algebra element $\nu \in \mathfrak{g}$, and parameterized by a single value $t \in \mathbb{R}$ often interpreted as time. A sequence-to-sequence model $\Phi$, mapping from $(f_0, \ldots, f_T) \mapsto (y_0, \ldots, y_T)$ is then said to be flow equivariant if, when the input sequence undergoes a flow, the output sequence also transforms according to the action of a flow, i.e.

$$\Phi\left(\{\psi_i(\nu) \cdot f_i\}_{i=0}^T\right)_t = \psi_t(\nu) \cdot \Phi\left(\{f_i\}_{i=0}^T\right)_t \ \forall t, \tag{1}$$

where the action of the flow on a signal $f_t$ over the group $G$ is defined as the left action: $\psi_t(\nu) \cdot f_t(g) := f_t(\psi_t(\nu)^{-1} \cdot g)$. To achieve flow equivariance, Keller (2025) demonstrated that it is sufficient to perform computation in the co-moving reference frame of the input. In other words, for a simple Recurrent Neural Network (RNN), the hidden state must flow in unison with the input, i.e.

$$h_{t+\Delta t} = \sigma\left(\psi_{\Delta t}(\nu) \cdot h_t + f_t\right). \tag{2}$$

To achieve equivariance with respect to a set of multiple flows ($\nu \in V$), Flow Equivariant RNNs possess multiple hidden state 'velocity channels', each flowing according to their own vector fields $\nu$ (denoted as $h_t(\nu)$), illustrated as stacked rows in Fig. 3 a). Because of the fact that the elements of the Lie algebra combine in a structured manner, it is then possible to show that when the input sequence is acted on by a flow $\psi(\hat{\nu})$, the hidden state outputs also flow, and these 'velocity channels' permute according to the difference between their velocity and the input velocity ($\nu - \hat{\nu}$):

$$h_t[\psi(\hat{\nu}) \cdot f](\nu) = \psi_{t-1}(\hat{\nu}) \cdot h_t[f](\nu - \hat{\nu}) \ \forall t. \tag{3}$$

In the following subsection, we will propose that in order to gain the efficiency and robustness benefits of equivariance in the world modeling setting, the 'hidden state' or memory of a world model can be group-structured with respect to both the group of the agent's actions, and the group which defines the abstract motions of other objects in the world. We can then act on this memory with the inverse of the representation of the agent's action in the output space ($T'^{-1}_{action}$), introducing equivariance of this memory with respect to self-motion, while the internal 'velocity channels' handle equivariance with respect to external motion. Fundamentally, this self-motion equivariance enforces the closure of group operations, such that if a set of actions brings an agent back to a previously observed location, the representation will necessarily be the same. As we will show empirically, this addresses the above described challenges of partially observed dynamic world modeling.

**Generalized Flow Equivariant Recurrence Relation.** To support more complex tasks, such as 3D partially observed world modeling, we introduce an abstract version of the flow equivariant recurrence relation which supports arbitrary encoders and update operations. Specifically, we define our abstract observation encoder as $E_\theta[f_t; h_t]$, a function of the current observation $f_t$ and the prior hidden state $h_t$; and we define our abstract recurrent update operation as $h_{t+1} = U_\theta[h_t; o_t]$, a function of the encoded observation ($o_t = E_\theta[f_t; h_t]$) and the past hidden state. Putting them together, the new *generalized flow equivariant recurrence relation* can then be written in as:

$$h_{t+1}(\nu) = \psi_1(\nu) \cdot U_\theta\big[h_t(\nu);\ E_\theta[f_t; h_t](\nu)\big] \tag{4}$$

To prove that this is indeed still flow equivariant, we require the following properties for the encoder and update mechanism. Specifically, both the encoder and update operations must be equivariant with respect to transformations on their inputs:

$$E_\theta[g \cdot f_t;\ g \cdot h_t] = g \cdot E_\theta[f_t; h_t] \quad \& \quad U_\theta[g \cdot h_t;\ g \cdot o_t] = g \cdot U_\theta[h_t; o_t] \tag{5}$$

Secondly, we also require that the Encoder performs a 'trivial lift' of the input to all velocity channels, such that: $E_\theta[f_t; h_t](\nu) = E_\theta[f_t; h_t](\hat\nu)\ \forall \nu, \hat\nu \in \mathfrak{g}$. In Appendix Section A, we prove formally that this framework indeed retains the flow equivariance properties of the original Flow Equivariant RNN, given that these assertions hold.

**Self-Motion Equivariance.** In this work, we leverage the fact that motion is relative (i.e. self-motion is equivalent to the motion of the input) to additionally achieve equivariance to self-motion in a unified manner – with the core difference being that self-motion is accompanied by a known action-variable ($a_t$) between the intervening observations. This additional information allows us to build a world model which operates in the co-moving reference frame of the agent, thereby achieving self-motion equivariance, without any additional 'velocity channels' – we call this model *FloWM*.

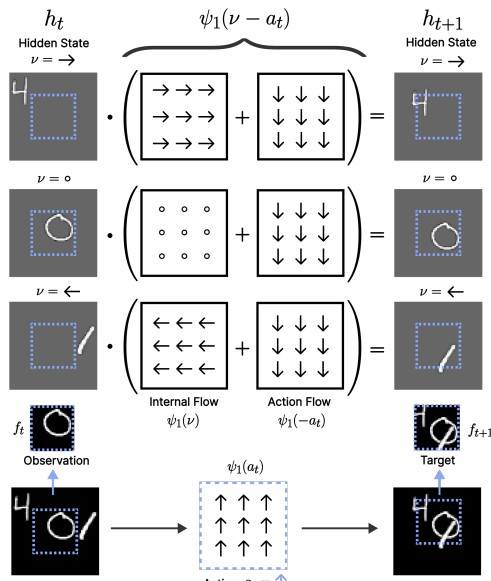

Figure 3: **Visualization of the Simple Recurrent FloWM on MNIST World.** FloWM Recurrence relation. Velocity channels are plotted as rows, with the 'read-in' and 'read-out' part of the hidden state in blue.

Specifically, given the action-variable $a_t$, denoting the action of the agent between observations $f_t$ and $f_{t+1}$, we transform the hidden state of the network to flow according to the latent group representation of the action. We assert the representation of the action on this hidden state is known, denoted $T_{a_t}$, resulting in the following *Self-Motion Flow Equivariant* Recurrence Relation:

$$h_{t+1}(\nu) = T_{a_t}^{-1} \psi_1(\nu) \cdot U_\theta\big[h_t(\nu);\ E_\theta[f_t, h_t](\nu)\big]. \tag{6}$$

In the case when the action space is the 2D translation group (such as in our MNIST World experiments in the following section), the representation $T_{a_t}$ takes the form of another 'Action Flow'

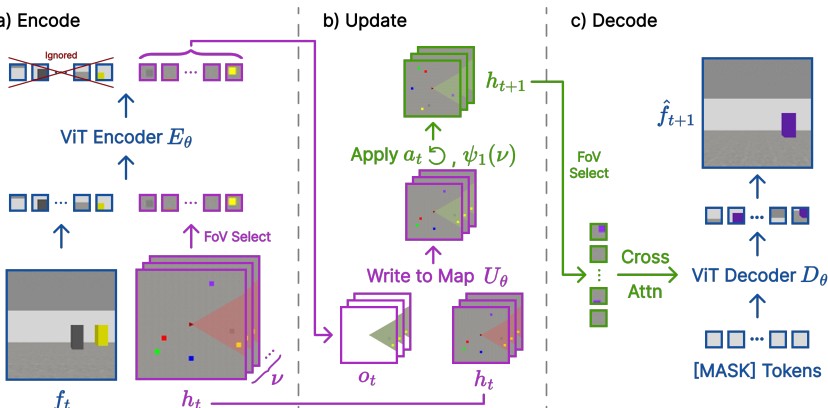

Figure 4: **Transformer-Based FloWM. a)** Image observation $f_t$ at time $t$ and FoV selected map latents $h_t$ are passed through ViT encoder $E_\theta$. Latent map $h_t$ is fully learned, visualized as a map here for clarity. **b)** Write to $h_t$ at the FoV locations, then transform latent map according to known action $a_t$ and internal flow $\psi_1(\nu)$, producing $h_{t+1}$. **c)** Decode using cross attention over FoV of $h_{t+1}$ with a ViT decoder $D_\theta$ to predict next image $\hat{f}_{t+1}$.

$(\psi_1(-a_t))$ describing the visual flow induced by the action in the agent's reference frame. By the properties of flows, this then combines with the 'Internal Flows' ($\psi_1(\nu)$) of the 'velocity channels', yielding a simple combined flow, $\psi_1(\nu - a_t)$, in the recurrence. When the action space is more sophisticated, such as involving rotations, the representation acts directly on the spatial dimensions and velocity channels of the hidden state itself. In the following paragraphs we describe precisely how these abstract elements are instantiated for each of the datasets we explore in this study.

## 3.2 Instantiations for 2D / 3D Partially Observed Dynamic World Modeling

**Simple Recurrent FloWM.** For the first set of experiments, to validate our framework in a 2D environment, we extend the model of Keller (2025) with the unified self-motion equivariance introduced above (depicted in Figure 3). Explicitly, this yields the following recurrence:

$$h_{t+1}(\nu) = \psi_1(\nu - a_t) \cdot \sigma\big(\mathcal{W} \star h_t(\nu) + \mathrm{pad}(\mathcal{U} \star f_t)\big), \qquad (7)$$

where $\mathcal{W} \star h_t$, and $\mathcal{U} \star f_t$ denote convolutions over the hidden state and input spatial dimensions. To model partial observability, we simply write-in-to (denoted 'pad($\cdot$)'), and read-out-from, a fixed $\mathrm{window\_size} < \mathrm{world\_size}$ portion of the hidden state (blue dashed square in Fig. 3(a)), letting the rest of the hidden state flow around the agent's field of view according to $\psi_1(\nu - a_t)$. In particular, the hidden state is windowed at each timestep, pixel-wise max-pooled over 'velocity channels' and passed through a decoder $g_\theta$ to predict the next observation, explicitly: $\hat{f}_{t+1} = g_\theta\big(\max_\nu(\mathrm{window}(h_{t+1}))\big)$. We see that this is equivalent to an instantiation of our general framework with $E_\theta[f_t; h_t](\nu) = \mathcal{U} \star f_t$, and $U_\theta[h_t; o_t] = \sigma(\mathcal{W} \star h_t + \mathrm{pad}(o_t))$ where all operations are equivariant to translation, and thus satisfy the conditions of Equation 5.

**Transformer-Based FloWM.** To extend our FloWM framework to work with more complex datasets, such as our second set of experiments involving a 3D world with other moving elements, we construct a second instantiation of the FloWM with a Vision Transformer (ViT) (Dosovitskiy et al., 2021) based encoder and decoder, depicted in Fig. 4. Specifically, in this setting $h_t$ is a set of spatially organized token embeddings that act as a group-structured latent map. In the spirit of Ha & Schmidhuber (2018), we set this map to be a 'top-down' 2D abstract version of the true 3-dimensional environment the agent inhabits. We denote this set of tokens $h_t := \{h_t^{(x,y)} | (x,y) \in [0,W) \times [0,H)\}$ where $(x,y)$ are the spatial coordinates of the token. Most importantly, this map is group-structured with respect to the agent's action group (2D translation and 90-degree rotation), and the group of external object motion (2D translation), giving us a known form of the representation of these group elements in the latent map, $T_g$.

The goal of the encoder $E_\theta[f_t; h_t]$, instantiated as a ViT, is then to take the tokens of the map corresponding to the current field of view, and update them using the image patch tokens ($\mathrm{patchify}(f_t)$).

Explicitly: $\mathrm{E}_\theta[f_t; h_t] = \mathrm{ViT}[\mathrm{concat}[\mathrm{patchify}(f_t); \mathrm{FoV}(h_t)]] = o_t$, where $\mathrm{FoV}(h_t)$ returns a fixed subset of $h_t$ corresponding to the 2D triangular wedge field of view of the agent, depicted in Fig. 4. We highlight that the coordinates of this set are fixed since the map is always *egocentric*, shifting and rotating around the agent in the center. The update operation $\mathrm{U}_\theta[h_t; o_t]$ then simply performs a gated combination of the output of the encoder (map token latent positions only), and the corresponding FoV tokens of the hidden state. Explicitly, matching our framework:

$$\mathrm{U}_\theta[h_t; o_t]^{(x,y)} = \begin{cases} (1 - \alpha) * h_t^{(x,y)} + \alpha * o_t^{(x,y)} & \text{if } x, y \in \mathrm{FoV} \\ h_t^{(x,y)} & \text{otherwise} \end{cases}, \tag{8}$$

where $\alpha = \sigma(\mathbf{W}\, \mathrm{concat}[h_t^{(x,y)}; o_t^{(x,y)}])$ for some learnable gating weights $\mathbf{W}$. We can see that the update operation $\mathrm{U}_\theta$ is indeed equivariant with respect to shifts or rotations of the spatial coordinates of its inputs, satisfying equation 5 for $\mathrm{U}_\theta$. However, since the encoder must map from the 3D first person point of view of the agent, to an abstract top-down map, it is highly non-trivial to make this transformation exactly action equivariant by design (without relying on explicit depth unprojection). Therefore, instead, we simply treat the output of the encoder as if it were equivariant in the recurrence relation, and anticipate that the transformation $T_{a_t}^{-1} \psi_1(\nu)$ between timesteps will encourage the encoder to learn to become equivariant, as has been demonstrated in prior work (Keller & Welling, 2022; Keurti et al., 2024). As we will demonstrate empirically in the following section, in practice this appears to hold. We provide more model details in Appendix E, and in Section 5 we review related models that have similarly structured representations with respect to self-motion, but may be seen as special cases of this framework without input-flow equivariance.

## 4 EXPERIMENTS

In this section we present our 2D and 3D partially observable dynamic world modeling benchmarks, and compare the proposed FloWM framework against state-of-the-art video diffusion world models and ablations of our FloWM. Our results on these datasets validate that unified self-motion and external flow equivariance are useful for modeling dynamics out of the field of view, whereas current world model formulations struggle due to their lack of unified memory and inability to model dynamics naturally.

### 4.1 DIFFUSION-BASED BASELINES

**Diffusion Forcing Transformer.** Due to its claims of long term consistency and flexible inference abilities, we chose a History-guided Diffusion Forcing training scheme as a baseline, using latent diffusion with a CogVideoX-style transformer backbone, which we will call here DFoT (Song et al., 2025; Chen et al., 2024; Yang et al., 2025). For each dataset, we first train a spatial downsampling VAE to encode video frames into a latent representation before being fed to diffusion model. Unlike FloWM recurrent models, and due to the diffusion forcing objective, during training for DFoT we make no distinction between observation and prediction frames, and train on length 70 sequences in the self-attention window. To condition on actions, we follow CogVideoX-style action conditioning: first we embed the action into the hidden dimension, then concatenate these additional tokens to the self attention window. During inference, we maintain 50 clean context frames while making predictions using the following 20 frames; the prediction frames begin at full noise and are gradually denoised until they are clean with the diffusion model, and then the sliding window advances by 20 frames. More details, discussion, and training settings for DFoT can be found in Appendix G.2.

**Diffusion Forcing State Space Model.** As a representative comparative work on memory-augmented video diffusion models, we compare against a recently proposed approach (Po et al., 2025) that integrates State Space Models (Dao & Gu, 2024) with block-wise SSM scanning for long-context world modeling, combined with a local frame-transformer attention window. This model employs the same VAE as DFoT across all datasets to operate in latent space. Training is performed on sequences of length 70, where the first 50 frames serve as clean context and the last 20 frames are noised according to the diffusion forcing objective, encouraging the model to leverage the clean context (Po et al., 2025). During inference, we follow the same practice as in DFoT and refer to this model as DFoT-SSM. Further details are provided in Appendix G.4.

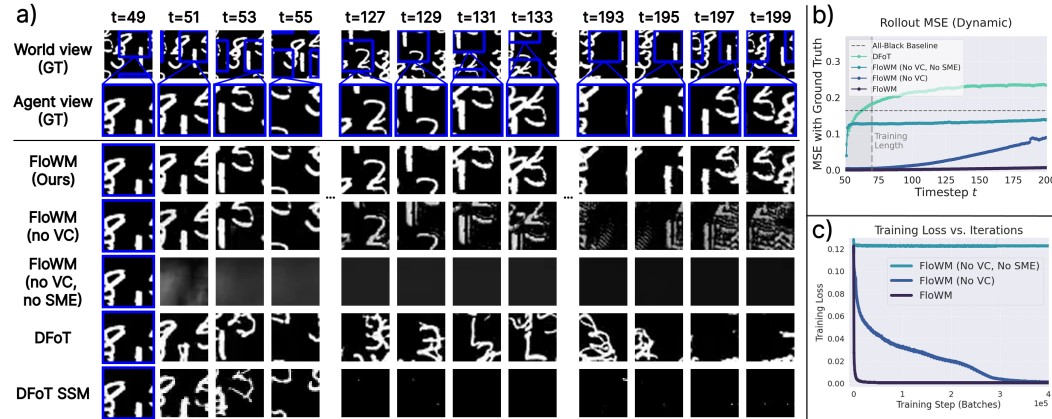

Figure 5: **Dynamic MNIST World Prediction Rollouts a)** Timesteps 0 to 49 are given as observations. Models are trained to predict up to $t = 70$. Note that FloWM does not diverge even at timestep 199, while baselines slowly degrade in image quality or lose track of the digits. **b)** MSE over different length rollouts show length generalization. **c)** Learning efficiency of the FloWM.

## 4.2 2D MNIST WORLD BENCHMARK

**Dataset** To test our architecture on partially observable dynamic world modeling, we propose a simple MNIST World dataset. The world is a 2D black canvas with multiple MNIST digits moving with random constant velocities. The agent is provided a view of the world, smaller than the world size, yielding partial observability. At each discrete timestep, the world evolves according to the velocity of each object, and the agent takes a random action (relative $(x, y)$ offset) to move its viewpoint. The edges roll, so a digit moving off the screen to the left will reappear on the right. Given 50 observation frames, the task is to predict the dynamics played out for 20 future frames, integrating future self-motion (given) and world dynamics. To test length generalization, each validation video has 50 observation frames and 150 prediction frames. We include ablations on data subsets with different combinations of partial observability, object dynamics, and self-motion in Appendix C.

**Results** On the MNIST World dataset, we train and evaluate the Simple Recurrent FloWM introduced in Section 3.2, which includes velocity channels (VC) and self-motion equivariance (SME). We also include ablations FloWM (no VC), FloWM (no VC, no SME), and the diffusion baselines mentioned above. We note here that FloWM (no VC, no SME) is just a simple convolutional RNN. More training and model details are available in Appendices E and G.2. At each timestep, we calculate various quality metrics for the predicted frames for each model with respect to the ground truth, reported in Table 1. Example rollouts and full world view visualizations are available in Fig. 5(a).

Predictions from the FloWM remain consistent with the motion of objects out of its view for 150 timesteps past the observation window, well beyond its training horizon of 20 prediction timesteps, while FloWM with (no VC, no SME) fails. We find that the FloWM with (no VC) can still somewhat learn to model unobserved dynamics, especially within its training window, but drifts over time; length extrapolation abilities are presented in Fig. 5(b). We further find models combining SME and VC require orders of magnitude less training steps to converge, shown in Fig. 5(c). The DFoT model's predictions quickly diverge from the ground truth, even within its training window, just generating plausible digit-like artifacts. The DFoT-SSM model's predictions show the digits slowly fading to black. Through additional results in Appendix C, we explore how the DFoT model can sometimes handle partial observability, object dynamics, and self-motion individually, but not in any combination.

## 4.3 3D DYNAMIC BLOCK WORLD BENCHMARK

**Dataset** Reasoning about the dynamics of the 3D world from 2D image observations requires approximating unprojection of egocentric views to a world-centric representation. To validate FloWM on this more difficult setting, we further introduce a simple 3D dataset, built in the Miniworld environment (Chevalier-Boisvert et al., 2023). An agent is spawned in a random position in a square room, along with colored blocks initialized with random positions and velocities. Each timestep,

Table 1: **Rollout performance on 2D Dynamic MNIST World with partial observability.** Models are evaluated by conditioning on 50 context frames, then generating 20 (training length) and 150 (length generalization) future frames respectively.

| Model | MSE ↓ | | PSNR ↑ | | SSIM ↑ | |
|---|---|---|---|---|---|---|
| | 20 | 150 | 20 | 150 | 20 | 150 |
| **FloWM (Ours)** | **0.0005** | **0.0018** | **32.99** | **27.56** | **0.9900** | **0.9813** |
| (no VC) | 0.0041 | 0.0334 | 23.83 | 14.77 | 0.9576 | 0.7729 |
| (no SME) | 0.1234 | 0.1317 | 9.088 | 8.805 | 0.0366 | 0.0127 |
| (no SME, no VC) | 0.1233 | 0.1333 | 9.091 | 8.751 | 0.0374 | 0.0146 |
| (+ action-concat) | 0.1125 | 0.1359 | 9.491 | 8.669 | 0.0623 | 0.0149 |
| DFoT | 0.1448 | 0.2111 | 8.394 | 6.755 | 0.4045 | 0.2434 |
| DFoT-SSM | 0.1277 | 0.1688 | 8.940 | 7.726 | 0.4550 | 0.3146 |
| All-Black Baseline | 0.1656 | 0.1654 | 7.810 | 7.814 | 0.3573 | 0.3564 |

Table 2: **Rollout performance on 3D Dynamic Block World.** Models are trained to generate 20 future frames given 50 context frames, and evaluated by generating 20 (training length) and 150 (length generalization) future frames respectively, conditioned on 50 context frames.

| Model | MSE ↓ | | PSNR ↑ | | SSIM ↑ | |
|---|---|---|---|---|---|---|
| | 20 | 150 | 20 | 150 | 20 | 150 |
| **FloWM (Ours)** | **0.00181** | **0.00279** | **27.43** | **25.54** | **0.9519** | **0.9442** |
| (no VC) | 0.00725 | 0.01071 | 21.40 | 19.70 | 0.9068 | 0.8945 |
| (no SME, no VC) | 0.01459 | 0.01760 | 18.36 | 17.54 | 0.8332 | 0.8285 |
| DFoT | 0.01001 | 0.01969 | 20.00 | 17.06 | 0.9496 | 0.9038 |
| DFoT-SSM | 0.01539 | 0.02171 | 18.13 | 16.63 | 0.9134 | 0.8854 |

the blocks evolve according to their velocities, and the agent takes one of four discrete actions: turn left, turn right, move straight, or do nothing. The blocks bounce when encountering a wall, making the task of modeling dynamics out of view significantly harder. The data-generating agent follows a biased random exploration strategy, sometimes pausing to observe the room's dynamics from next to the wall. Videos of the agent's observations are collected; following the MNIST World setup, the world model is given 50 observation frames as context and must predict the dynamics evolving over the next 20 steps, given the agent's actions. More details, including ablations involving a static version of Block World are included in Appendix B, and more training details are available in Appendix E.

**Results** On the 3D Dynamic Block World dataset, we compare our Transformer-Based FloWM from Section 3.2 and Fig. 4 with the diffusion baseline, also including the ablations FloWM (no VC) and FloWM (no VC, no SME). We report the metrics on rollouts of 20 and 150 frames in Table 2. Example rollouts are visualized in Figure 6 and on the website here. Similar to the 2D experiments, we observe the FloWM's predictions are able to remain consistent up to 150 frames of future prediction, while the baselines and ablations are not. Perceptually, DFoT and SSM model predictions frequently hallucinate new objects and forget old ones, aligning with the hypothesis that their architectures are not well suited for partially observable dynamic environments. In addition to the standard 3D Dynamic Block World dataset, we evaluate our model and baselines on a significantly more visually difficult Textured 3D Dynamic Block World dataset, where block, wall, and floor textures are randomly assigned for each example. The same gap holds, demonstrating that FloWM is additionally applicable to more visually realistic settings. Results for this set are available in Appendix B.

## 5 RELATED WORK

**Generative World Modeling with Memory.** As mentioned in Section 2, there have been a few prior works that augment diffusion video models to improve long horizon consistency and go beyond the limited context window of transformer-based diffusion models. Here, we will discuss the ones most relevant to this work in detail, and leave broader related work to Appendix F. To begin, all work mentioned here focuses on static scenes only, and conditions on actions through token-based conditioning, lacking any way to harmoniously integrate the agent's actions over time. The baseline

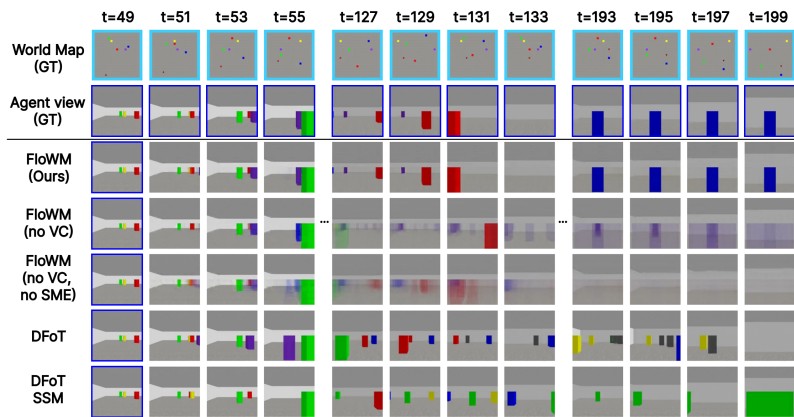

Figure 6: **Dynamic Block World Prediction Rollouts.** Timesteps 0 to 49 are given as observations. Note that FloWM stays consistent until the final frame, while the baselines hallucinate object position and color. The ablations are unable to remain consistent.

DFoT-SSM model incorporates a recurrent SSM backbone for the purposes of long-context consistency and memory; however, the SSM is primarily remembering viewpoint-dependent observations, similar to extending the self-attention context window, and does not have any explicit computation steps for predicting the state of the world out of the view of the agent (Po et al., 2025; Dao & Gu, 2024). Another recently developed, separate model, WORLDMEM augments diffusion video models by placing past image observations in a memory bank for later retrieval based on the camera position (Xiao et al., 2025). An existing ablation of their model in their paper involving camera view prediction may encourage the model to learn how to integrate its global position in time, similar to the effect of self-motion equivariance in our work, but their memory mechanism is fundamentally different, relying on self-attention to integrate information for self-consistency. Recent work from Zhou et al. (2025) maintains a 3d voxel map of the environment, later retrieved to condition diffusion generation. However, their method relies on depth unprojection, the voxels are updated via max pooling instead of a more flexible recurrence relation, and are prohibitively expensive for large hidden state sizes.

**Equivariant World Modeling.** Perhaps most related to our proposed FloWM are world modeling frameworks with similarly structured 'map'-memories. For example, Neural Map (Parisotto & Salakhutdinov, 2017) introduced a spatially organized 2D memory that stores observations at estimated agent coordinates. The storage location of these observations is shifted precisely according to the agent's actions, yielding an effectively equivariant 'allocentric' latent map. In Section 5 of the paper, the authors describe an egocentric version of their model which can in fact be seen as a special case of our FloWM, specifically equivalent to the ablation without velocity channels. The authors demonstrate that their allocentric map enables long-term recall and generalization in navigation tasks. In a similar vein, EgoMap (Beeching et al., 2020) leverages inverse perspective transformations to map from observations in 3D environments to a top-down egocentric map. This work also explicitly transforms the latent map in an action-conditioned manner, although the transformation is learned with a Spatial Transformer Network, making it only approximately equivariant. Our work can be seen to formalize these early models in the framework of group theory, allowing us to extend the action space beyond just spatial translation to any Lie group and any world space. For example, our framework can theoretically support full 3-dimensional 'neural maps' without problem, following the framework of flow equivariance. Finally, there are a few other works that discuss equivariant world modeling, but are less precisely related to our own. Specifically, (van der Pol et al., 2021) was one of the first works to build equivariant policy and value networks for reinforcement learning, but not with respect to motion, instead with respect to the symmetries of the environment (such as static rotations or translations). More recent work (Park et al., 2022; Ghaemi et al., 2025) proposes to approach the goal of building equivariant world models in a more approximate manner by conditioning or encouraging equivariance through training losses, rather than our approach which builds it in explicitly.

## 6 LIMITATIONS & FUTURE WORK.

Our current instantiations of FloWM target environments where both the agent and objects undergo relatively simple, rigid motions under a known action parameterization. This setting lets us isolate the role of the flow equivariant memory under partial observability, but it does not yet capture richer non-rigid or semantic actions (e.g. articulated bodies, deforming objects, or discrete semantic actions such as "open door" or "pick up object"). Extending the flow equivariant recurrence to actions that live in more expressive latent groups or hierarchies, and to domains where agent actions are semantic rather than purely geometric, is an important direction for future work.

Our experiments also focus on deterministic dynamics given an action sequence, and we train FloWM with a single-step reconstruction loss to predict a single future rollout. We view the ability to model deterministic trajectories as a prerequisite to eventually model stochastic ones. The same flow equivariant latent map could in principle be combined with stochastic latent variables to enable stochastic dynamic prediction, representing another interesting line of future work.

Training and inference compute requirements for FloWM remain within the same order of magnitude as the strongest baselines despite maintaining a recurrent state. We report compute metrics and suggest several future directions to further improve efficiency in Appendix H.

There are a few architectural limitations to note in this current study. As noted, our 3D FloWM ViT Encoder is not analytically equivariant with respect to 3D transformations. While the model appears to learn this equivariance over time, and therefore still benefits from the group-structured hidden state, we observe that this results in slower learning initially until a point where approximate equivariance appears learned, and the model can leverage the velocity channels properly. Future work that incorporates a proper analytically equivariant 3D encoder would likely observe significantly faster training speeds and lower loss, akin to the performance we report on the 2D dataset. Second, Flow Equivariance to date has only been developed with respect to discrete sets of flows $V$, while real world velocities may span a continuous range. However, prior work has repeatedly demonstrated empirically that even equivariance to small discretized groups yields performance improvements on data that is symmetric with respect to the full group (Cohen & Welling, 2016; Kuipers & Bekkers, 2023). Recent theoretical work has further characterized the value of such 'approximate' or 'partial' group equivariance, demonstrating the value that such methods hold even if not exact (Petrache & Trivedi, 2025). Future work to extend flow equivariance to continuous velocities would regardless still hold significant value. Lastly, our latent world map is egocentric and fixed in spatial extent and resolution. While the textured BlockWorld results indicate that FloWM can handle increased visual complexity, scaling to more realistic, open world scenes will likely require variable sized maps and stronger perceptual backbones.

Finally, we believe a complementary line of work is to combine FloWM with insights from non-generative world models. In this paper, we refer to "embodied" as an egocentric sensory stream paired with self motion actions, and evaluate FloWM purely as a predictive world model. In the future, the flow equivariant latent memory that we introduce in this paper could be used as a representation backbone within JEPA or TDMPC2 style world model to improve long horizon dynamics predictions (LeCun & Courant, 2022; Hansen et al., 2024). Together, they could represent a major step forward to tackle downstream embodied tasks such as autonomous driving, robotic manipulation, or game environments through planning over the model predictions or latent space.

## 7 CONCLUSION

In this work, we have introduced Flow Equivariant World Models, a new framework unifying both internally and externally generated motion for more accurate and efficient world modeling in partially observable settings with dynamic objects out of the agent's view. Our results on both the 2D and 3D datasets with these properties demonstrate the potential of flow equivariance, and highlight the limitations with current state-of-the-art diffusion-based video world models. Specifically, we find that flow equivariant world models are able to represent motion in a structured symmetric manner, permitting faster learning, lower error, fewer hallucinations, and more stable rollouts far beyond the training length. We believe this work lays the theoretical groundwork along with empirical validation to support the potential for a novel symmetry-structured approach for efficient and effective world modeling.

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

# A    GENERALIZED FLOW EQUIVARIANCE PROOF

In this section, we prove by induction that the generalized Flow Equivariant Recurrence Relation of Equation 4 is indeed flow equivariant, following the proof technique of Keller (2025).

First, to restate the problem, we wish to show that for the recurrence defined as follows:

$$h_{t+1}(\nu) = \psi_1(\nu) \cdot U_\theta \big[ h_t(\nu); \; E_\theta \left[ f_t, h_t \right] (\nu) \big], \tag{9}$$

if we assume that:

1. The encoder satisfies the 'trivial lift' condition to the velocity field: $E_\theta \left[ f_t; h_t \right] (\nu) = E_\theta \left[ f_t; h_t \right] (\hat{\nu}) \;\; \forall \nu, \hat{\nu} \in \mathfrak{g}$

2. The encoder and decoder are both group equivariant with respect to their arguments: $E_\theta \left[ g \cdot f_t; \; g \cdot h_t \right] = g \cdot E_\theta \left[ f_t; h_t \right] \quad \& \quad U_\theta \left[ g \cdot h_t; \; g \cdot o_t \right] = g \cdot U_\theta \left[ h_t; \; o_t \right]$

3. The hidden state is initialized to be constant along the flow dimension and invariant to the flow action: $h_0(\nu) = h_0(\nu') \, \forall \nu', \nu \in V$ and $\psi_1(\nu) \cdot h_0(\nu) = h_0(\nu) \, \forall \nu \in V$,

then, the follow flow equivariance commutation relation holds:

$$h_t[\psi(\hat{\nu}) \cdot f](\nu) = \psi_t(\hat{\nu}) \cdot h_t[f](\nu - \hat{\nu}) \;\; \forall t. \tag{10}$$

*Theorem* (The Generalized Flow Equivariant Recurrence Relation of Eqn. 9 is Flow Equivariant). Let $h[f] \in \mathcal{F}_{K'}(Y, \mathbb{Z})$ be a the output of the generalized flow equivariant recurrence relation as defined in Equation 9, with hidden-state initialization invariant to the group action and constant in the flow dimension, i.e. $h_0(\nu, g) = h_0(\nu', g) \, \forall \nu', \nu \in V$ and $\psi_1(\nu) \cdot h_0(\nu, g) = h_0(\nu, g) \, \forall \nu \in V, g \in G$. Then, $h[f]$ is flow equivariant with the following representation of the action of the flow in the output space for $t \geq 1$:

$$(\psi(\hat{\nu}) \cdot h[f])_t(\nu, g) = h_t[f](\nu - \hat{\nu}, \psi_t(\hat{\nu})^{-1} \cdot g) \tag{11}$$

We note for the sake of completeness, that this then implies the following equivariance relations:

$$h_t[\psi(\hat{\nu}) \cdot f](\nu, g) = h_t[f](\nu - \hat{\nu}, \psi_t(\hat{\nu})^{-1} \cdot g) = \psi_t(\hat{\nu}) \cdot h_t[f](\nu - \hat{\nu}, g) \tag{12}$$

Furthermore, in all settings in this work, $G$ refers to the 2D translation group, either indexing pixel coordinates (MNIST), or 2D latent map coordinates (Block world) referred to as $(x, y)$ in the main text.

We see that, different from the work of Keller (2025), since the generalized recurrence relation in Eqn. 9 applies the flow update after the input has been combined with the hidden state (i.e. outside the $U_\theta$ operator), the commutation relation in Eqn. 10 now has a $t$ index on $\psi_t$, instead of $t - 1$ as written in equation 3.

*Proof.* (Theorem, Generalized Flow Equivariance)

Base Case: The base case is trivially true from the initial condition:

$$h_0[\psi(\hat{\nu}) \cdot f_{<0}](\nu, g) = h_0[f_{<0}](\nu, g) \quad \text{(by initial cond. being independent of input)} \tag{13}$$

$$= h_0[f_{<0}](\nu - \hat{\nu}, \psi_t(\hat{\nu})^{-1} \cdot g) \quad \text{(by constant init.)} \tag{14}$$

Inductive Step: Assuming $h_t[\psi(\hat{\nu}) \cdot f](\nu, g) = \psi_t(\hat{\nu}) \cdot h_t[f](\nu - \hat{\nu}, g) \, \forall \nu \in V, g \in G$, for some $t \geq 0$, we wish to prove this also holds for $t + 1$:

Using the Generalized Flow Recurrence (Eqn. 9) on the transformed input, we get:

$$h_{t+1}[\psi(\hat{\nu})\cdot f](\nu, g) = \psi_1(\nu)\cdot U_\theta\Big(h_t[\psi(\hat{\nu})\cdot f](\nu, g) \; ; \; E_\theta[(\psi_t(\hat{\nu})\cdot f_t),\; h_t[\psi(\hat{\nu})\cdot f]](\nu)\Big) \quad (15)$$

$$\text{(by inductive hyp.)} \quad = \psi_1(\nu)\cdot U_\theta\Big(\psi_t(\hat{\nu})\cdot h_t[f](\nu - \hat{\nu}, g) \; ; \; E_\theta[(\psi_t(\hat{\nu})\cdot f_t),\; \psi(\hat{\nu})_t\cdot h_t[f]](\nu)\Big) \quad (16)$$

$$\text{(trivial lift in } \nu) \quad = \psi_1(\nu)\cdot U_\theta\Big(\psi_t(\hat{\nu})\cdot h_t[f](\nu - \hat{\nu}, g) \; ; \; E_\theta[(\psi_t(\hat{\nu})\cdot f_t),\; \psi(\hat{\nu})_t\cdot h_t[f]](\nu - \hat{\nu})\Big) \quad (17)$$

$$\text{(equivariance of } E_\theta) \quad = \psi_1(\nu)\cdot U_\theta\Big(\psi_t(\hat{\nu})\cdot h_t[f](\nu - \hat{\nu}, g) \; ; \; \psi_t(\hat{\nu})\cdot E_\theta[f_t,\; h_t[f]](\nu - \hat{\nu})\Big) \quad (18)$$

$$\text{(equivariance of } U_\theta) \quad = \psi_1(\nu)\cdot\psi_t(\hat{\nu})\cdot U_\theta\Big(h_t[f](\nu - \hat{\nu}, g) \; ; \; E_\theta[f_t,\; h_t[f]](\nu - \hat{\nu})\Big) \quad (19)$$

$$\text{(flow composition)} \quad = \psi_{t+1}(\hat{\nu})\cdot\psi_1(\nu - \hat{\nu})\cdot U_\theta\Big(h_t[f](\nu - \hat{\nu}, g) \; ; \; E_\theta[f_t,\; h_t[f]](\nu - \hat{\nu})\Big) \quad (20)$$

$$\text{(by Eqn. 9)} \quad = \psi_{t+1}(\hat{\nu})\cdot h_{t+1}[f](\nu - \hat{\nu}, g) \quad (21)$$

$$= h_{t+1}[f]\big(\nu - \hat{\nu},\; \psi_{t+1}(\hat{\nu})^{-1}\cdot g\big). \quad (22)$$

Thus, assuming the inductive hypothesis for time $t$ implies the desired relation at time $t+1$; together with the base case this completes the induction and proves the Theorem. $\qquad\square$

## B  BLOCK WORLD ADDITIONAL DATASET DETAILS AND RESULTS

### B.1  BLOCK WORLD DATASET DETAILS

Here, we describe dataset generation and parameter settings for the Dynamic (presented in the main text), Textured Dynamic, and Static subsets of Block World. The dataset generation parameters are described in Table 3. Each dataset example has a video of shape `[num_frames, channels, height, width]`, where channels is 3 for RGB; and an accompanying actions list of shape `[num_frames]`, for the discrete actions taken by the agent: left, right, forward, or do nothing. Left and right correspond to a 90 degree rotation such that the agent stays aligned with the grid. Each dataset subset contains 10,000 videos for training, and 1,000 videos for validation. The world size describes the number of coordinates in the world able to be occupied by the agent or a block. The agent is spawned in a random location within the world.

The Dynamic Block World dataset subset has blocks with randomized colors chosen from among blue, green, yellow, red, purple, and gray. There are no collisions between the blocks, or between

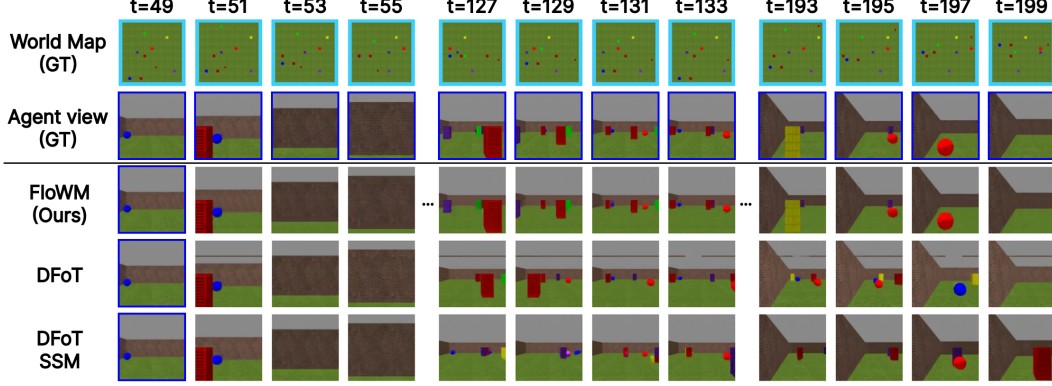

Figure 7: **Textured 3D Block World Rollouts.** Here we visualize additional qualitative rollout results on the Textured 3D Block World split. Note the hallucinated objects in both DFoT and DFoT-SSM as time goes on, whereas FloWM remains consistent.

Table 3: Generation parameters for Block World dataset subsets.

| Data Subset | World Size | Observation Resolution | # Blocks | Block Velocity Range, x and y |
|---|---|---|---|---|
| Static Block World | 15x15 | 128x128 | 6-10 | 0 |
| Dynamic Block World | 15x15 | 128x128 | 6-10 | -1 to +1 |
| Textured Dynamic Block World | 15 | 128x128 | 6-10 | -1 to +1 |

Table 4: **Rollout results on Textured 3D Dynamic Block World.** Models are trained to generate 90 future frames given 50 context frames, and evaluated by generating 150 future frames conditioned on 50 context frames.

| Model | MSE $\downarrow$ | PSNR $\uparrow$ | SSIM $\uparrow$ |
|---|---|---|---|
| **FloWM (Ours)** | **0.00129** | **28.88** | **0.9342** |
| DFoT | 0.00888 | 20.52 | 0.8588 |
| SSM | 0.01081 | 19.66 | 0.8532 |

the agent and the blocks, but the blocks bounce off the wall and change direction, making the motion nonlinear.

The textured dataset has the same dynamics behavior as the Dynamic Block World dataset, but with the following randomizations: (i) textures are randomized for the wall, chosen between brick, dark wood panel, wood panel; (ii) textures are randomized for the floor, chosen between cardboard, grass, concrete; (iii) textures are randomized for the blocks, chosen between metal grill, airduct grate, cinder blocks, and ceiling tiles; they remain colored randomly; (iv) 1/3 of the time, the object is a sphere instead of a block. In combination, these all add significant visual complexity and randomness to the environment.

The static subset just has the velocity of the objects initialized to 0; the agent's exploration pattern is the same. We used Miniworld for this environment, and would like to thank the authors and contributors of Miniworld for creating a flexible and useful 3d simulator environment for agents (Chevalier-Boisvert et al., 2023).

### B.2 TEXTURED BLOCK WORLD RESULTS

In this section we report additional results on the Textured Dynamic Block World dataset described above. The focus of this work is on the introduction of the flow equivariant framework for modeling partially observable dynamics, rather than the strength of the visual encoder / decoder. However the fact that FloWM can retain its performance improvements over the baselines in this settings suggests it can serve as a step forward for a general framework capable of encoding visually realistic scenes as well.

The results are available in Table 4. Example rollouts are visible on the project website here. Frames of the rollouts are visible in Figure 7. The metric numbers are not easily comparable to the other dataset split due to the different dataset statistics, but the relative distance is still clearly noticeable.

### B.3 STATIC BLOCK WORLD RESULTS

In this section we report additional results on the static Block World dataset described above. Results are available in Table 5. As with the static MNIST World dataset, in this setting, the default configuration of FloWM with velocity channels only adds noise to the model, since the velocity channels have no external motion to model. Therefore, it is unsurprising that the FloWM (no VC) achieves the best metric scores. Despite the environment being static, DFoT and DFoT-SSM still struggle with keeping consistent with information that may have left their immediate context window. We hypothesize that due to the baseline models' lack of a spatial memory, combined with the randomized number of blocks per environment, these models are unable to consistently remember where the blocks are.

Table 5: **Rollout performance on 3D Static Block World with partial observability.** Models are trained to generate 20 future frames given 50 context frames, and evaluated by generating 20 (training length) and 150 (length generalization) future frames respectively, conditioned on 50 context frames.

| Model | MSE ↓ | | PSNR ↑ | | SSIM ↑ | |
|---|---|---|---|---|---|---|
| | 20 | 150 | 20 | 150 | 20 | 150 |
| **FloWM (Ours)** | 0.00274 | 0.00697 | 25.62 | 21.57 | 0.7839 | 0.7691 |
| (no VC) | **0.00113** | **0.00179** | **29.48** | **27.47** | **0.9763** | **0.9730** |
| (no SME, no VC) | 0.01619 | 0.02091 | 17.91 | 16.80 | 0.8055 | 0.8036 |
| DFoT | 0.00591 | 0.01784 | 22.28 | 17.49 | 0.9653 | 0.9107 |
| SSM | 0.00824 | 0.02173 | 20.84 | 16.63 | 0.9485 | 0.8878 |

# C  MNIST WORLD ADDITIONAL DATASET DETAILS AND RESULTS

## C.1  MNIST WORLD DATASET DETAILS

| Data Subset | Self-Motion | Dynamics | Partially Observable |
|---|---|---|---|
| dynamic_fo_no_sm | No | Yes | No |
| dynamic_fo | Yes | Yes | No |
| static_po | Yes | No | Yes |
| dynamic_po | Yes | Yes | Yes |

Table 6: MNIST world data subsets demonstrating scaling difficulty in self-motion, dynamics, and partial observability.

Here, we describe dataset generation and parameter settings for our ablations on self-motion, dynamics, and partial observability in the MNIST World setting. The subsets are succinctly described in Table 6, and the generation parameters in Table 7. A subset is described as partially observable if the world size is larger than the window size. We also scale the number of digits by the size of the world. Each dataset example has a video of shape [num_frames, channels, height, width], where channels is 1; and an accompanying actions list of shape [num_frames, 2], for the x and y translation of the agent view at each timestep. The dynamic_fo_no_sm subset just has dynamics and is fully observable; the dynamic_fo subset has dynamics and is fully observable, but also has self-motion; the static_po subset is partially observable, and the agent has self-motion, but the digits do not move; and finally, the dynamic_po subset includes partial observability, agent movement, and dynamics. In the main text, we report all results on just the dynamic_po subset. For all subsets with dynamics, each digit is given an integer velocity for x and y in the digit velocity range (e.g., -2 to 2). For each dataset subset with self-motion, at each step during the observation and prediction phase, a random integer is chosen in x and y to be the agent's view translation, bounded by the self-motion range (e.g., -10 to 10). For each dataset, objects that move across the boundary reappear on the other side as a circular pad. Each dataset subset contains 180,000 videos in the training set, and 8,000 videos in the validation set. Results on each of these data subsets for FloWM and baseline models are described below.

| Data Subset | World Size | Window Size | # Digits | Self-motion Range | Digit Velocity Range, x and y |
|---|---|---|---|---|---|
| dynamic_fo_no_sm | 32 | 32 | 3 | 0 | -2 to +2 |
| dynamic_fo | 32 | 32 | 3 | 10 | -2 to +2 |
| static_po | 50 | 32 | 5 | 10 | 0 |
| dynamic_po | 50 | 32 | 5 | 10 | -2 to +2 |

Table 7: Generation parameters for MNIST World dataset subsets.

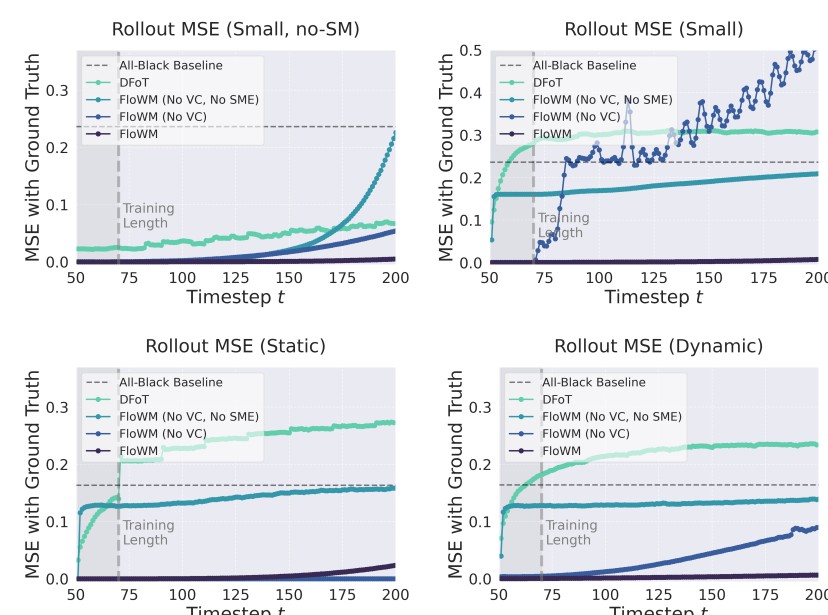

Figure 8: Rollout Error (MSE) vs. Forward Prediction Steps for all data subsets of MNIST World. The dynamic subset is replicated from the main text for ease of comparison.

### C.2 MNIST WORLD ADDITIONAL RESULTS

Here we report additional results on the MNIST World subsets described above. We evaluate FloWM and FloWM ablations described in Appendix E. We compare to the DFoT and DFoT-SSM baselines as described in Appendix G.4.

The error (MSE) between the predicted future observations (rollout) and the ground truth is plotted for each baseline in Figure 8 as a function of forward prediction timestep (x-axis). Metrics are reported over the first 20 timesteps (the training length) and over the full 150 timesteps (length generalization) in Tables 9, 8, 10. Due to being constructed with a different number of digits, metrics between the data subsets are not necessarily directly comparable. We provide the All-Black Baseline (model that only predicts **0** for future observations) as a form of normalization for comparison.

All models are able to do reasonably well on the simplest fully observable dataset with no self-motion; note here the DFoT is doing latent diffusion, so there is a small amount of error error from the decoding step, contributing a baseline MSE of around 0.02, see Appendix G.5 for more details. This setup aligns with the typical setting of world modeling, where the information that the model needs is expected to be in the attention window. The other dataset splits do not follow this assumption, and the results align with expectations about the model's capabilities. The DFoT does relatively better on the static `static_po` compared to the `dynamic_po` dataset, due to not having to model dynamics, but the model's outputs still diverge from the ground truth quickly.

For a dataset where the velocity channels are redundant, i.e. `static_po`, the FloWM (no VC) does slightly better than FloWM. Further note that the FloWM (no VC) is able to have low error on most of the tasks, though with a much higher value than FloWM as errors accumulate due to not having the velocity channels to encode flow equivariantly. Taken together, the ablations suggest that self-motion equivariance is key to solving the problem, and that input flow equivariance via velocity channels helps with exactness and convergence time, with the tradeoff of a larger hidden state activation size.

Table 8: **Rollout performance on 2D Dynamic MNIST World with full observability while without self-motion.** Models are trained to generate 20 future frames given 50 context frames, and evaluated by generating 20 (training length) and 150 (length generalization) future frames respectively, conditioned on 50 context frames.

| Model | MSE ↓ | | PSNR ↑ | | SSIM ↑ | |
|---|---|---|---|---|---|---|
| | 20 | 150 | 20 | 150 | 20 | 150 |
| **FloWM (Ours)** | 0.00009 | **0.00104** | 40.61 | 29.83 | **0.9982** | 0.9752 |
| (no SME) | **0.00006** | 0.00173 | **41.89** | 27.61 | 0.9981 | **0.9797** |
| (no SME, no VC) | 0.00031 | 0.02845 | 35.05 | 15.46 | 0.9950 | 0.7746 |
| (no VC) | 0.00026 | 0.00925 | 35.87 | 20.34 | 0.9877 | 0.9061 |
| (action concat) | 0.00015 | 0.01349 | 38.36 | 18.70 | 0.9973 | 0.8099 |
| DFoT | 0.02128 | 0.03601 | 16.72 | 14.44 | 0.8961 | 0.8205 |
| SSM | 0.01298 | 0.22098 | 18.87 | 6.56 | 0.9399 | 0.2907 |

Table 9: **Rollout performance on 2D Dynamic MNIST World with full observability.** Models are trained to generate 20 future frames given 50 context frames, and evaluated by generating 20 (training length) and 150 (length generalization) future frames respectively, conditioned on 50 context frames.

| Model | MSE ↓ | | PSNR ↑ | | SSIM ↑ | |
|---|---|---|---|---|---|---|
| | 20 | 150 | 20 | 150 | 20 | 150 |
| **FloWM (Ours)** | 0.00013 | **0.00136** | 39.01 | **28.65** | 0.9977 | **0.9763** |
| (no SME) | 0.15595 | 0.16167 | 8.07 | 7.91 | 0.0355 | 0.0105 |
| (no SME, no VC) | 0.15597 | 0.18251 | 8.07 | 7.39 | 0.0351 | 0.0141 |
| (no VC) | **0.00010** | 0.26278 | **40.21** | 5.80 | **0.9982** | 0.1718 |
| (action concat) | 0.13425 | 0.15148 | 8.72 | 8.20 | 0.0796 | 0.0294 |
| DFoT | 0.22194 | 0.29699 | 6.54 | 5.27 | 0.2086 | 0.0833 |
| SSM | 0.08162 | 0.25271 | 10.88 | 5.97 | 0.6170 | 0.1090 |

# D  FLOWM EXPERIMENT DETAILS: 3D DYNAMIC BLOCK WORLD

On the 3D Dynamic Block World Dataset, FloWM is built with 6-layer ViT encoders and decoders with 8 attention heads per layer, and an embedding dimension of 256.

## D.1  RECURRENCE

The hidden state $h_t \in \mathbb{R}^{|V| \times C_{hid} \times H_{world} \times W_{world}}$ has $|V|$ velocity channels (indexed by the elements $\nu \in V$), and $C_{hid} = 256$ hidden state channels (the same as the ViT token embedding dimension). The spatial dimensions of the hidden state are set to match the world size for each dataset, meaning $H_{world} = W_{world} = 16$. The hidden state is initialized to all zeros for the first timestep, i.e. $h_0 = \mathbf{0}$.

## D.2  VELOCITY CHANNELS

On the 3D Block World dataset, we add velocity channels only up to $\pm 1$ in both the X and Y dimensions of the image with no diagonal velocities. Thus in total, $|V| = 5$ for the FloWM. Each channel is flowed by its corresponding velocity field (defined by $\psi(\nu)$) at each step. This is denoted by $\psi_1(\nu) \cdot h_t(\nu)$.

The actions of the agent then induce an additional flow of the hidden state, which we implement via the inverse of the representation of the action $T_{-a_t} = T_{a_t}^{-1}$. In practice, this is implemented by performing a roll operation on the hidden state by exactly 1 element for a forward action, and a $+/-$ 90-degree rotation for left or right actions respectively.

## D.3  TRANSFORMER DETAILS

The ViT Encoder takes in both image input tokens and the FoV-selected map latent tokens, and processes them together in the self-attention window. We use a patch size of 16 to patchify the image before a sin-cos absolute position embedding is added (Vaswani et al., 2023). The map latent

Table 10: **Rollout performance on 2D Static MNIST World with partial observability.** Models are trained to generate 20 future frames given 50 context frames, and evaluated by generating 20 (training length) and 150 (length generalization) future frames respectively, conditioned on 50 context frames.

| Model | MSE ↓ | | PSNR ↑ | | SSIM ↑ | |
|---|---|---|---|---|---|---|
| | 20 | 150 | 20 | 150 | 20 | 150 |
| **FloWM (Ours)** | 0.00013 | 0.00467 | 38.78 | 23.31 | 0.9970 | 0.8053 |
| (no SME) | 0.12097 | 0.12790 | 9.17 | 8.93 | 0.0460 | 0.0116 |
| (no SME, no VC) | 0.12086 | 0.13046 | 9.18 | 8.85 | 0.0520 | 0.0144 |
| (no VC) | **0.00002** | **0.00004** | **47.63** | **44.49** | **0.9994** | **0.9993** |
| (action concat) | 0.09473 | 0.11860 | 10.24 | 9.26 | 0.1595 | 0.1354 |
| DFoT | 0.10505 | 0.22641 | 9.79 | 6.45 | 0.5199 | 0.2257 |
| SSM | 0.05398 | 0.15706 | 12.68 | 8.04 | 0.7274 | 0.3397 |

Table 11: Block World ViT FloWM Configurations.

| Component | Option | Value |
|---|---|---|
| Training | Learning rate | 1e-4 |
| | Effective batch size | 16 |
| | Training steps | 150k |
| | GPU usage | 2×H100 |
| Model | Hidden channels | 256 |
| | Encoder depth | 6 |
| | Encoder heads | 8 |
| | Decoder dim | 256 |
| | Decoder depth | 6 |
| | Decoder heads | 8 |
| | Patch size | 16 |
| | N Params | 10M |
| | Patch size | 16 |

tokens are also added together with a sin-cos absolute position embedding based on the position in the 2d map.

### D.4 TRAINING DETAILS

To train the FloWM, as well as the ablated versions, we provide the model with 50 observation frames as input, and train the model to predict the next 20 observations conditioned on the corresponding action sequence. Specifically, we minimize the mean squared error (MSE) between the output of the model and the ground truth sequence, averaged over the 20 frames (from frame 50 to 70):

$$\mathcal{L}_{MSE} = \frac{1}{20} \sum_{t=50}^{70} ||f_t - \hat{f}_t||_2^2. \tag{23}$$

The models are trained with the Adam optimizer with a learning rate of $1e-4$, a batch size of 16. They are each trained for 150k steps, or until converged.

## E    FLOWM EXPERIMENT DETAILS: MNIST WORLD

In this work, we introduce the Flow Equivariant World Model (FloWM). The Simple Recurrent version of this model, tested on the 2D MNIST World Dataset, is built as a simple sequence-to-sequence RNN with small CNN encoders/decoders to model MNIST digit features. Full code is available on the project page.

For completeness, we repeat the Simple Recurrent Simple Recurrent FloWM recurrence relation below:

$$h_{t+1}(\nu) = \sigma\big(\psi_1(\nu - a_t) \cdot \mathcal{W} \star h_t(\nu) + \mathrm{pad}(\mathcal{U} \star f_t)\big). \tag{24}$$

### E.1 RECURRENCE

The hidden state $h_t \in \mathbb{R}^{|V| \times C_{hid} \times H_{world} \times W_{world}}$ has $|V|$ velocity channels (indexed by the elements $\nu \in V$), and $C_{hid} = 64$ hidden state channels. The spatial dimensions of the hidden state are set to match the world size for each dataset. For the partially observed world, this means $H_{world} = W_{world} = 50$ (where the window size is set to $32 \times 32$), while for the fully observed world, $H_{world} = W_{world} = 32$. The hidden state is initialized to all zeros for the first timestep, i.e. $h_0 = \mathbf{0}$.

The hidden state is processed between timesteps by a convolutional kernel $\mathcal{W}$. This kernel has the potential to span between velocity channels, and therefore model acceleration or more complex dynamics than static velocities. In this work, since our dataset has no such dynamics (we only have constant object velocities), we safely ignore the inter-velocity convolution terms, and simply set $\mathcal{W}$ to be a $3 \times 3$ convolutional kernel, with 64 input and output channels, circular padding, and no bias. We refer the interested reader to Keller (2025) for details on the form of the full flow-equivariant convolution that could be equally used in this model. The hidden state is finally passed through a non-linearity $\sigma$ to complete the update to the next timestep. In this work, for MNIST World, we use a ReLU.

### E.2 VELOCITY CHANNELS

In this work, for MNIST World, we add velocity channels up to $\pm 2$ in both the X and Y dimensions of the image. Explicitly, $V = \{(-2, -2), (-2, -1), \ldots (0, 0) \ldots (2, 2)\}$. Thus in total, $|V| = 25$ for the FloWM. Each channel is flowed by its corresponding velocity field (defined by $\psi(\nu)$) at each step. This is denoted by $\psi_1(\nu) \cdot h_t(\nu)$.

The actions of the agent then induce an additional flow of the visual stimulus. In order to be equivariant with respect to this flow in addition to the flows in $V$, we simply additionally flow each hidden state by the corresponding inverse of the action flow $\psi(-a_t)$. In total this gives the combined flow for each flow channel $\psi_1(\nu - a_t)$. In practice, this is implemented by performing a roll operation on the hidden state by exactly $(\nu - a_t)$ pixels.

### E.3 ENCODER

The 'encoder' is simply a single convolutional layer, with $3 \times 3$ kernel $\mathcal{U}$, 1 input channel, and 64 output channels. The convolution uses circular padding, and no bias. The observation at timestep $t$ ($f_t$), is thus processed by the encoder ($\mathcal{U} \star f_t$) yielding the processed observation of the agent. Given this observation is only a partial observation of the full world, we must pad this observation to match the world size, and the size of the hidden state. We denote this operation as 'pad' in the recurrence relation, and simply pad the boundary of the output of the encoder with 0 to match the world-size (size of the hidden state).

### E.4 DECODER

We learn the parameters of the FloWM by training it to predict future observations from its hidden state and the corresponding sequence of future actions. To compute this prediction, we take a consistent window_size crop from the center of the hidden state, corresponding to the same location where the encoder 'writes-in'. We denote this crop window($h_{t+1}$). To then enable the model to predict each pixel's velocity independently, we perform a pixel-wise max-pool over the $V$ dimension ('velocity channels') before passing the result to a decoder $g_\theta$. Specifically: $\hat{f}_{t+1} = g_\theta\left(\max_\nu \left(\text{window}(h_{t+1})\right)\right)$. The decoder $g_\theta$ is a simple 2 layer convolutional neural network with $3 \times 3$ convolutional kernels, 64 hidden channels, and a ReLU non-linearity between the layers.

### E.5 ABLATION: NO VELOCITY CHANNELS

To construct the ablated version of the FloWM with no velocity channels, we simply set $V = \{(0, 0)\}$. Since the original FloWM model simply max-pools over velocity channels, the decoder already only takes a single velocity channel as input, so no other portions of the model need to change. We note that this model is identical to a simple convolutional recurrent neural network with

self-motion equivariance. Explicitly:

$$h_{t+1} = \sigma\big(\psi_1(-a_t) \cdot \mathcal{W} \star h_t + \text{pad}(\mathcal{U} \star f_t)\big). \tag{25}$$

### E.6 Ablation: No Self-Motion Equivariance

To construct the ablated version of the FloWM with no self-motion equivariance, we simply remove the term $-a_t$ from the flow of the recurrence relation. Explicitly:

$$h_{t+1}(\nu) = \sigma\big(\psi_1(\nu) \cdot \mathcal{W} \star h_t(\nu) + \text{pad}(\mathcal{U} \star f_t)\big). \tag{26}$$

We note that this is equivalent to the original FERNN model with the addition of the partial-observability modifications (padding the input and windowing the hidden state for readout).

### E.7 Ablation: No Velocity Channels + No Self-Motion Equivariance

To ablate both velocity channels and self-motion equivariance, we reach a simple convolutional RNN:

$$h_{t+1} = \sigma\big(\mathcal{W} \star h_t + \text{pad}(\mathcal{U} \star f_t)\big). \tag{27}$$

### E.8 Ablation: Conv-RNN + Action Concat

In the appendix, we additionally include a version of the model with no velocity channels, no self-motion equivariance, but with action conditioning for both the input and hidden state. Specifically, we concatenate the current action to the hidden state vector and the input image as two additional channels (corresponding to the x and y components of the action translation vector), and change the number of input channels for both convolutions correspondingly. Explicitly:

$$h_{t+1} = \sigma\big(\mathcal{W} \star \text{concat}(h_t, a_t) + \text{pad}(\mathcal{U} \star \text{concat}(f_t, a_t))\big). \tag{28}$$

Empirically, we find that this additional conditioning marginally improves the model performance; however, the model is still clearly unable to learn the precise equivariance that the FloWM has built-in.

### E.9 Training Details

To train the FloWM, as well as the ablated versions, we provide the model with 50 observation frames as input, and train the model to predict the next 20 observations conditioned on the corresponding action sequence. Specifically, we minimize the mean squared error (MSE) between the output of the model and the ground truth sequence, averaged over the 20 frames (from frame 50 to 70):

$$\mathcal{L}_{MSE} = \frac{1}{20} \sum_{t=50}^{70} ||f_t - \hat{f}_t||_2^2. \tag{29}$$

The models are trained with the Adam optimizer with a learning rate of $1e-4$, a batch size of 32, and gradient clipping by norm with a value of $1.0$. They are each trained for 50 epochs, or until converged. Some models, such as the FloWM with self-motion equivariance but no velocity channels, took longer than 50 epochs to converge, and thus training was extended to 100 epochs. All FloWM models (and ablations) have roughly 75K trainable parameters.

## F Additional Related Work

**World Modeling** Generative World Models (Ha & Schmidhuber, 2018; Brooks et al., 2024) aim to simulate and predict how environments evolve over time. They have broad applications in reinforcement learning (Hafner et al., 2023; Samsami et al., 2024; Hansen et al., 2024), autonomous driving (Bar et al., 2025; Yang et al., 2023; Russell et al., 2025; Jia et al., 2023; Wang et al., 2024), robotics (Yang et al., 2023; Agarwal et al., 2025; Ali et al., 2025; Taniguchi et al., 2023), and planning (Hao et al., 2023; Cloos et al., 2024), where agents must anticipate future states in order to reason and act (Gao et al., 2025). Recent work has moved toward building general large-scale interactive world simulators with persistent states (Ball et al., 2025; Agarwal et al., 2025; Xiang et al.,

2024; He et al., 2025; Dynamics Lab, 2025; Guo et al., 2025). However, a critical limitation remains: most generative world models lack any semblance of memory. Autoregressive rollouts with transformer models have limited context windows, and remaining consistent with information outside of the context window requires something beyond naive self attention. As a result, long rollouts often produce contradictions with earlier context or previously generated sequences, undermining temporal coherence. Although recent explorations have introduced memory mechanisms through explicit (with 3D priors) (Zhou et al., 2025; Xiao et al., 2025; Yu et al., 2025) or implicit (based on neural / learned components) (Po et al., 2025; Gu et al., 2025; Savov et al., 2025) memory mechanisms, the focus has been on maintaining physical consistency with a static world. Dynamics are included in some datasets, but are not the main focus; especially the dynamics of objects outside the current field of view has not yet been studied as the main focus with the context of image space world modeling, to our knowledge. Prediction of the world necessarily requires predicting the state evolution of occluded or unseen entities, especially in real-world settings; we thus believe addressing this gap is crucial for advancing toward faithful and embodied generative world models.

**Partially Observable Environments and Tasks** Partial observability is a fundamental challenge in embodied AI and reinforcement learning, and a variety of benchmarks and environments have been developed to study it. Existing tasks can be broadly categorized along three dimensions. The first concerns whether the underlying environment is dynamic or static. The second concerns how partial observability (PO) is introduced: (i) PO within a fixed view, typically through object occlusion or objects moving in and out of view; (ii) PO outside the current view, which requires changes in the ego perspective. The third dimension is the task objective: some benchmarks are designed for question answering under PO, while others target next-state prediction, i.e. world modeling. We posit that the combination of all three dimensions – dynamic environments, with partial observability outside the current view, targeting world modeling – is currently understudied and underdeveloped.

A first branch of existing work lies in the fixed-view setting, where PO arises from occlusion in 3D scenes. Vision benchmarks such as CATER (Girdhar & Ramanan, 2020) and IntPhys (Riochet et al., 2020) capture such occlusion scenarios, but remain focused on passive observation without involving an embodied agent or dynamics outside the current field of view.

A second branch tests out-of-view partial observability, frequently utilizing simulators to control the state of the world. Early work introducing partially observable markov decision processes in environments like partially observable pacman require a rough understanding of the state of the world that may include dynamics (Hausknecht & Stone, 2017), but the task is entangled with decision making on a limited domain. 3D platforms such as DMLab (Beattie et al., 2016), VizDoom (Kempka et al., 2016), and MiniWorld/MiniGrid (Chevalier-Boisvert et al., 2023), probe exploration or combat with compact observations. For example, Pasukonis et al. (2022) introduces a memory maze using DMLab to evaluate the static memory of RL agents. Among these simulators, DMLab is largely static, while VizDoom and MiniGrid can introduce dynamics out-of-view.

Newer benchmarks approach dynamic PO but for the most part, they are primarily focused on QA rather than explicit next-state forecasting. Dynamic House (Kurenkov et al., 2023) includes evolving scenes and asks for future relations (e.g., whether an object moved rooms when hidden), effectively casting dynamics as link prediction. Hazard Challenge (Zhou et al., 2024) evaluates decision-making under evolving disasters (e.g. fire, flood). WorldPredictions (Chen et al., 2025) studies world modeling in a partially observable semi-MDP, but emphasizes action selection and procedural planning. Similarly, first-person embodied QA datasets over long videos (Ye et al., 2025; Das et al., 2018; Kim & Ammanabrolu, 2025) combine partial observability with dynamics but are retrospective: they query past observations rather than test predictive reasoning about where previously dynamic objects might be now.

Although many relevant tasks and environments exist, to our knowledge, there are still no direct evaluations designed to test world models on their ability to understand, encode, and predict dynamics under partial observability. Nevertheless, these existing environments provide valuable building blocks for constructing benchmarks that assess whether generative world models can move beyond static memory to capture the evolving dynamics of objects out of view.

**Equivariant World Models** Equivariant models respect the symmetry of their data, ensuring that structured transformations in the input induce predictable changes in the model's internal state. This

inductive bias has indeed been found to be valuable in prior work on world modeling. Specifically, although not explicitly framed as equivariant, one of the most related world modeling architectures to our proposed self-motion equivariance is the Neural Map (Parisotto & Salakhutdinov, 2017). This work introduced a spatially organized 2D memory that stores observations at estimated agent coordinates. The storage location of these observations is shifted precisely according to the agent's actions, yielding an effectively equivariant 'allocentric' latent map. In Section 5 of the paper, the authors describe an egocentric version of their model which can in fact be seen as a special case of our FloWM, specifically equivalent to the ablation without velocity channels. The authors demonstrate that their allocentric map enables long-term recall and generalization in navigation tasks. In a similar vein, EgoMap (Beeching et al., 2020) built on this by leveraging inverse perspective transformations to map from observations in 3D environments to a top-down egocentric map. This work also explicitly transforms the latent map in an action-conditioned manner, although the transformation is learned with a Spatial Transformer Network, making it only approximately equivariant. Our work can be seen to formalize these early models in the framework of group theory, allowing us to extend the action space beyond just spatial translation to any Lie group and any world space. For example, our framework can theoretically support full 3-dimensional 'neural maps' without problem, following the framework of flow equivariance. Finally, there are a few other works that discuss equivariant world modeling, but are less precisely related to our own. Specifically, (van der Pol et al., 2021) was one of the first works to build equivariant policy and value networks for reinforcement learning, but not with respect to motion, instead with respect to the symmetries of the environment (such as static rotations or translations). More recent work (Park et al., 2022; Ghaemi et al., 2025) proposes to approach the goal of building equivariant world models in a more approximate manner by conditioning or encouraging equivariance through training losses, rather than our approach which builds it in explicitly. Overall, we find all of these approaches to be complementary to our own and are excited for their combined potential.

**Neuroscience.** Excitingly, in the neuroscience literature, there is evidence for predictive processing in the mammalian visual system which is a function of self-motion signals. For example, Keller et al. (2012) and Leinweber et al. (2017) have found that responses in visual cortex are strongly modulated by self-motion signals, and mismatch between predicted and experienced stimuli. Similarly, it is known that position coding in the hippocampus through place cells and grid cells forms an equivariant map through phase coding of agent location. Explicitly, the phase of a given place cell's spike shifts *equivariantly* with respect to the agent's forward motion along a linear track. Further computational and theoretical work has demonstrated that grid-like activations emerge automatically through the enforcement of equivariance in cell responses Dorrell et al. (2023). We believe this work suggests that there may indeed be a biological mechanism for encouraging the visual system to be a G-equivariant map from stimuli to a type of latent G-space.

## G  Baseline Details

### G.1  Video Diffusion Transformers

Diffusion Transformer based video generation models are the most prominent so-called world models today (Peebles & Xie, 2023; Brooks et al., 2024). Training follows a similar formula with diffusion image generation pipelines, requiring attention over the temporal dimension to retain temporal consistency. For video data, diffusion models are typically trained within the latent space of a variational autoencoder (VAE) (Rombach et al., 2022; Gupta et al., 2023), where raw video frames are first compressed into compact latent representation.

The ability for these video diffusion models to generate impressively realistic videos has led to an increased interest for their use as world models, and there is a growing focus in ensuring the spatiotemporal consistency of these models as world simulators. Due to the size complexity of the input token space, to generate long videos, researchers have turned to autoregressive sampling and sliding window attention; though ubiquitously used, we speculate that the drawbacks of this method for inference, where there is no hidden state passed between generation rounds after the window shifts, is a major reason that DiT baselines fails on the simple task presented in this work.

## G.2 DIFFUSION FORCING TRANSFORMER BASELINE

Due to its claims of long term consistency and flexible inference abilities, for our baseline we chose a History-guided Diffusion Forcing training scheme, using latent diffusion with a CogVideoX-style transformer backbone, which we will call here DFoT (Song et al., 2025; Chen et al., 2024; Yang et al., 2025; Rombach et al., 2022). Models for state of the art video world modeling today have similar training formulas and architectures for the backbone (Xiang et al., 2024; Ball et al., 2025; Decart et al., 2024; Agarwal et al., 2025). We first trained a spatial downsampling VAE on frames of the MNIST-world data subsets, then pass input video frames through the VAE to form a latent representation before it reaches the diffusion model.

Following the standard diffusion forcing training scheme, each frame during training is corrupted with independent gaussian noise, and the training target is to predict some form of the ground truth from these noisy frames. Song et al. (2025) showed that using this training schedule allows for the history image frames to be prepended to the noisy frames as context in the same self attention window, with zero (or some minimal) noise level, called History Guidance.

For DFoT models, unlike FloWM recurrent models, during training we make no distinction between observation and prediction frames, and train on length 70 sequences in the self-attention window, where each frame's tokens receive independent gaussian noise. During inference, we utilize History Guidance with 70 frames in the attention window to provide image context for consistent generation. Specifically, the 50 observation frames are given minimal noise, and the 20 prediction frames all begin at full noise; then the entire set of frames is passed through the model multiple times according to the scheduler to complete denoising the target frames to get clean frames as outputs. Specifically, each latent frame in the sequence $\mathbf{x}_t \in \mathbf{x}_\tau$ is assigned an independent noise level $k_t \in [0, 1]$. Each frame (more precisely, each collection of spatial tokens corresponding to a single frame) is noised according to the following equation:

$$\mathbf{x}_t^{k_t} = \alpha_{k_t}\mathbf{x}_t^0 + \sigma_{k_t}\epsilon_t, \quad \epsilon_t \sim \mathcal{N}(0, 1), \tag{30}$$

where $\alpha_{k_t}$ and $\sigma_{k_t}$ denote the signal and noise scaling factors, respectively, determined by the chosen variance schedule. The diffusion model $\epsilon_\theta$ takes in as input a sequence of noise levels, $k_\tau$, and the sequence of independently noised inputs $\mathbf{x}_\tau^{k_\tau}$. The model is trained to minimize the following diffusion loss:

$$\mathbb{E}_{k_\tau, \mathbf{x}_\tau, \epsilon_\tau}\left[\left\|\epsilon_\tau - \epsilon_\theta\left(\mathbf{x}_\tau^{k_\tau}, k_\tau\right)\right\|^2\right]. \tag{31}$$

For more information on diffusion models in general, please see Chan (2025).

To run inference for generation of longer videos (as in the length extrapolation experiments), we use a sliding window approach, matching the number of frames seen during training in the self-attention window. Specifically, we keep 50 context frames, shift the window ahead by 20 frames after each chunk is done denoising, and use the newly generated frames as context for the next generation round.

## G.3 DFoT TRAINING DETAILS

We train a separate DFoT model for each MNIST World data subset to separate out its abilities. We embed actions using a simple MLP embedder, and concatenate it to the video tokens, following CogVideoX. Our 96M parameter DFoT's validation loss and validation metrics converge after 240k steps on 1 NVIDIA L40S 48GB GPU with a batch size of 128 on MNIST World, and after 300k steps on 1 NVIDIA L40S 48GB GPU with a batch size of 32 on Block World. More training hyperparameters are reported in Table 12.

## G.4 DFoT-SSM TRAINING DETAILS

We train a separate DFoT-SSM model for each data subset to separate out its abilities. We embed actions using a simple MLP embedder, and concatenate it to the video tokens, following CogVideoX. Our 97.8M parameter DFoT-SSM's validation loss and validation metrics converge after 200k steps on 2 NVIDIA L40S 48GB GPU with an effective batch size of 128 on MnistWorld, and 300k steps on 2 NVIDIA L40S 48GB GPU with an effective batch size of 32 on BlockWorld. More training hyperparameters are reported in Table 13.

Table 12: DFoT configurations for different datasets. Section and key are organized hierarchically in the first column.

| Config | MnistWorld | BlockWorld |
|---|---|---|
| **Training** | | |
| Effective batch size | 128 | 32 |
| Learning rate | 2e-4 (linear warmup) | 2e-4 (linear warmup) |
| Warmup steps | 2,000 | 2,000 |
| Weight decay | 1e-3 | 1e-3 |
| Training steps | 245k | 300k |
| GPU usage | 1×L40S | 2×L40S |
| Optimizer | Adam, betas=(0.9, 0.99) | Adam, betas=(0.9, 0.99) |
| Training strategy | Distributed Data Parallel | Distributed Data Parallel |
| Precision | Bfloat16 | Bfloat16 |
| **Diffusion** | | |
| Objective | $v$-prediction | $v$-prediction |
| Sampling steps | 50 | 50 |
| Noise schedule | cosine | cosine |
| Loss weighting | sigmoid | sigmoid |
| **Model** | | |
| Total parameters | 95.3 M | 95.3 M |
| # attention heads | 12 | 12 |
| Head dimension | 64 | 64 |
| # layers | 10 | 10 |
| Time embed dimension | 256 | 256 |
| Condition embed dimension | 768 | 768 |
| **Inference** | | |
| History guidance | stabilized conditional (level = 0.02) | stabilized conditional (level = 0.02) |
| Context frames | 50 | 50 |
| Sampler | DDIM | DDIM |

## G.5 VAE TRAINING DETAILS

Following standard practice, we use a VAE to perform latent diffusion; doing diffusion on pixels instead could offer perceptually different results, but we do not believe it would alter the results of the model. We train our 8x spatial downsampling VAE on sample frames from a mix of all of the data subsets, such that all combinations of overlapping MNIST digits are within the training distribution. Our 20M parameter VAE's validation loss converges at about 90k steps for MNIST World, using an effective batch size of 256 across 4 NVIDIA L40S 48GB GPUs with a learning rate of 4e-4. We utilize a Masked Autoencoder Vision Transformer based VAE (He et al., 2021). We directly apply the VAE code from Oasis (Decart et al., 2024), including an additional discriminator loss that helps with visual quality; please refer to their work for more details. The reconstruction MSE accuracy reaches 0.02 for MNIST World, so any DFoT MSE can be expected to be 0.02 higher than if trained on pixels; we believe this should not affect convergence behavior of the DFoT models on the downstream task. During diffusion training, for our VAE with latent dimension 4, and spatial downsampling ratio 8, input videos of shape `[num_frames, channels, height, width]` are converted to shape `[num_frames, 4, height // 8, width // 8]`. For Block World, the reconstruction error is less than 0.003, and we train the model for 300k steps with a latent dimension 8 and spatial downsampling ratio 16. More training hyperparameters are reported in Table 14 and Table 15.

Table 13: DFoT-SSM configurations. Classifier-free guidance (Ho & Salimans, 2022) for conditions is not used during inference; though the models have been trained to allow for it, we find their instruction following ability not to be a limiting factor. Loss weighting uses sigmoid reweighting proposed by Kingma & Gao (2023) and adopted by Hoogeboom et al. (2024). History guidance follows the stabilized conditional method (level = 0.02) from Song et al. (2025); please refer to their code base for details.

| Config | MnistWorld | BlockWorld |
|---|---|---|
| **Training** | | |
| Effective batch size | 128 | 32 |
| Learning rate | 2e-4 (linear warmup) | 2e-4 (linear warmup) |
| Warmup steps | 2,000 | 2,000 |
| Weight decay | 1e-3 | 1e-3 |
| Training steps | 200k | 300k |
| GPU usage | 1×L40S | 2×L40S |
| Optimizer | Adam, betas=(0.9, 0.99) | Adam, betas=(0.9, 0.99) |
| Training strategy | Distributed Data Parallel | Distributed Data Parallel |
| Precision | Bfloat16 | Bfloat16 |
| **Diffusion** | | |
| Objective | $v$-prediction | $v$-prediction |
| Sampling steps | 50 | 50 |
| Noise schedule | cosine | cosine |
| Loss weighting | sigmoid | sigmoid |
| **Model** | | |
| Total parameters | 97.8 M | 97.8 M |
| # attention heads | 12 | 12 |
| Head dimension | 64 | 64 |
| # layers | 10 | 10 |
| Time embed dimension | 256 | 256 |
| Condition embed dimension | 768 | 768 |
| **Inference** | | |
| History guidance | stabilized conditional (level = 0.02) | stabilized conditional (level = 0.02) |
| Context frames | 50 | 50 |
| Sampler | DDIM | DDIM |

# H    COMPUTE RESOURCE COMPARISON

## H.1    TRAINING AND INFERENCE COMPUTE COMPARISONS

To contextualize the computational footprint of FloWM, we report training compute (in EFLOPs) and inference wall-clock time for FloWM and our baselines. All measurements are obtained on a single NVIDIA H100 GPU. Training compute (reported in Table 16) is estimated by first measuring the forward and backward GFLOPs for a batch size of 1, and then scaling by the actual batch size and number of optimization steps. Under this common protocol, FloWM requires roughly 1.7 to $2.6\times$ more training FLOPs than DFoT and DFoT-SSM to reach convergence, but remains within the same order of magnitude while delivering substantially more stable long-horizon predictions.

Inference wall clock time is measured as the per-step latency for predicting a single frame with batch size 1 (reported in Table 17). These per-frame runtimes are also of the same order of magnitude across all models. Unlike fully feedforward architectures, FloWM maintains a recurrent world state. Therefore it cannot be parallelized over sequence length in its current form, and inference cost grows linearly with the number of predicted frames. We view this linear dependence as a deliberate tradeoff in exchange for preserving a coherent latent map over long horizons, which appears important for accurately representing hidden dynamics under partial observability. In our view, this

Table 14: VAE configurations for MNIST World. The input size from the dataset is $32 \times 32$.

| Component | Option | Value |
|---|---|---|
| Training | Learning rate | 4e-4 |
| | Effective batch size | 256 |
| | Precision | Float16 mixed precision |
| | Strategy | Distributed Data Parallel |
| | Warmup steps | 10,000 |
| | Training epochs | 172 |
| | GPU usage | 4×L40S |
| | Optimizer (AE) | Adam, betas=(0.5, 0.9) |
| | Optimizer (Disc) | Adam, betas=(0.5, 0.9) |
| Model | Total parameters | 19.7 M |
| | Encoder dim | 384 |
| | Encoder depth | 4 |
| | Encoder heads | 12 |
| | Decoder dim | 384 |
| | Decoder depth | 7 |
| | Decoder heads | 12 |
| | Patch size | 8 |
| Latent | Latent dim | 4 |
| | Temporal downsample | 1 |

Table 15: VAE configurations for Block World. The input size from the dataset is $128 \times 128$.

| Component | Option | Value |
|---|---|---|
| Training | Learning rate | 4e-4 |
| | Effective batch size | 256 |
| | Precision | Float16 mixed precision |
| | Strategy | Distributed Data Parallel |
| | Warmup steps | 10,000 |
| | Training epochs | 40 |
| | GPU usage | 4×L40S |
| | Optimizer (AE) | Adam, betas=(0.5, 0.9) |
| | Optimizer (Disc) | Adam, betas=(0.5, 0.9) |
| Model | Total parameters | 80.7 M |
| | Encoder dim | 576 |
| | Encoder depth | 5 |
| | Encoder heads | 12 |
| | Decoder dim | 576 |
| | Decoder depth | 15 |
| | Decoder heads | 12 |
| | Patch size | 16 |
| Latent | Latent dim | 8 |
| | Temporal downsample | 1 |

computational tradeoff is well worth it in order to attain the capability to predict the dynamics of the world consistently.

## H.2 LIMITATIONS AND OPPORTUNITIES FOR EFFICIENCY

Our current implementation is not heavily optimized, and there are several possible future avenues to reduce computational cost without changing the core modeling assumptions. First, decoupling the encoder from the hidden state update would allow processing of input frames in parallel before writing to the latent map, increasing input level parallelism. Second, designing the recurrent update to be linear and associative would, in combination with such a decoupling, make it amenable to parallel scan algorithms similar to those used in modern state-space models Martin & Cundy (2018); Smith et al. (2023). This would enable parallelization over sequence length while retaining a recur-

Table 16: **Training compute on 3D Dynamic BlockWorld.** FLOPs are estimated from per-sample forward/backward GFLOPs measured at batch size 1 and 140 frames on 1 H100 GPU, then scaled by the actual batch size and number of training steps.

| Model | Batch size | Steps | Total compute (EFLOPs) |
|---|---|---|---|
| FloWM (Ours) | 16 | 150k | 27.5 |
| DFoT | 32 | 300k | 15.7 |
| SSM | 32 | 300k | 10.5 |

Table 17: Per step compute and runtime on Block World with batch size 1 and 2 frames on 1 H100 GPU.

| Model | Forward GFLOPs | Backward GFLOPs | Forward Time (ms) | Backward Time (ms) |
|---|---|---|---|---|
| FloWM (Ours) | 31.18 | 99.50 | $104.47 \pm 101.05$ | $72.35 \pm 2.37$ |
| DFoT | 4.94 | 9.91 | $90.93 \pm 109.59$ | $106.94 \pm 4.70$ |
| SSM | 5.00 | 10.03 | $47.52 \pm 0.85$ | $89.85 \pm 0.42$ |

rent structure. Third, the latent map is currently updated via a dense gated operation over the entire estimated field of view, even though the true change in information per timestep is typically sparse. Introducing sparse or multiscale updates could substantially reduce per-step computation while preserving the benefits of a persistent world memory. Implementation could be inspired by other hierarchical mapping methods such as NICE-SLAM and SHINE-Mapping (Zhu et al., 2022; Zhong et al., 2023). Overall, these directions suggest that the additional compute required by FloWM is not fundamental to the architecture, but rather reflects design choices that can be systematically optimized in future work.

# I  LLM USAGE

In this work, we occasionally used LLMs for polishing writing, including for sentence rewriting suggestions and finding word synonyms.

