# OpenReview forum: "Flow Equivariant World Modeling for Partially Observed Dynamic Environments"
_ICLR.cc/2026/Conference — Submitted to ICLR 2026_

### Official Review · Reviewer_wAFH · 2025-10-28

**Soundness:** 3
**Presentation:** 3
**Contribution:** 3
**Rating:** 4
**Confidence:** 3

**Summary:**

This paper introduces Flow Equivariant World Models (FloWM), a novel framework for modeling dynamic environments that are only partially observable from an embodied agent’s egocentric viewpoint. The core idea is to unify self-motion and external object motion under the mathematical framework of Lie group flows, treating both as time-parameterized symmetries. By enforcing flow equivariance, the model maintains a structured, group-equivariant latent memory that evolves consistently with both the agent’s actions and the dynamics of unseen objects. The authors also propose a practical and scalable Transformer-based implementation of FloWM, and the experiments on 2D and 3D world modeling tasks demonstrate the effectiveness of FloWM.

**Strengths:**

- This paper addresses the challenges of partially-observed environments and memory limitations in world modeling, which are critical and challenging issues that previous approaches struggle to handle effectively.
- FloWM leverages the concept of flow equivariance and introduces two novel techniques: velocity channels and self-motion equivariance. These techniques elegantly unify both internal and external object motions in world modeling.
- The theoretical framework is rigorous, providing solid evidence for the effectiveness of FloWM's foundation.
- The authors also present a practical implementation of FloWM based on Transformer architecture, demonstrating its impressive performance on both 2D and 3D world modeling benchmarks. Ablation studies further validate the contribution of each component.

**Weaknesses:**

- FloWM assumes that both agents and objects move at a constant velocity along fixed trajectories, a strong assumption that limits its applicability to more complex tasks where motion is more variable and dynamic.
- FloWM relies on maintaining latent maps during both training and inference, which makes it less suitable for open-ended scenarios where the environment is constantly evolving and expanding.
- FloWM focuses primarily on the movement of agents and objects, but does not account for interactions between the agents and objects, which is an important aspect in embodied AI.
- Although the paper claims to present an **embodied** world model, the experimental evaluation is confined to simulated games rather than physical robotics tasks.

**Questions:**

- How does FloWM model non-uniform motion of agents and objects, especially when their speeds are variable or unpredictable?
- How can FloWM be applied to open-ended tasks where the task map is constantly evolving or expanding over time?
- How does FloWM account for interactions between the agent and objects, which are crucial for embodied AI?
- What is FloWM's performance in more complex physical embodied tasks, such as robotic manipulation or real-world applications?

I will raise my score if these questions are addressed well.

---

> ### Author Response · Authors · 2025-11-22
> **Official Response to Reviewer wAFH (Part 1)**
>
> ## Summary
>
> We would like to thank the reviewer for their thoughtful and positive evaluation. In particular, we are glad the reviewer found FloWM’s unified treatment of self motion and object motion via flow equivariance to be a theoretically grounded and empirically supported approach to the critical challenges of partial observability and memory in world modeling. Below, we address the reviewer's concerns, including:
>
> - Refining the motion assumptions in our model (W1, Q1)
> - Addressing the benefit of persistent latent maps for evolving and expanding environments (W2, Q2)
> - Clarifying our view on embodiment and environment interaction (W3, W4, Q3)
> - Introducing an additional experiment to assuage scalability concerns (Q4)
>
> ## Clarifying the motion assumptions (W1, Q1)
>
> We appreciate the reviewer for understanding the setting within which our model operates, and identifying a real challenge for representing variable dynamics in realistic datasets. We would like to emphasize that we chose environments with controlled, predictable dynamics in order to isolate the challenges of partial observability and long horizon memory and to cleanly compare the capabilities of different models. The main goal of this work is to show that even within this relatively simple regime of predictable dynamics, strong sliding window baselines (DFoT, DFoT-SSM), fail to maintain consistent long horizon predictions, whereas our architecture is able to maintain and update this map as time goes on.
>
> We also note that even in our 3D Blockworld experiments, object velocity is not strictly constant: when objects bounce off walls, their velocities change. These dynamics remain deterministic but must be inferred and predicted out of view, which our model handles via its persistent world map. Predicting dynamics in environments with deterministic action sequences can actually be viewed as a subset of all possible action sequences, and given the results of our baselines, this is already a major challenge. We thus see FloWM, which is successful in these deterministic regimes, as a step forward to eventually being able to predict stochastic dynamics and multiple futures.
>
> ## Latent maps for evolving environments (W2, Q2)
>
> We see the persistent latent map as a core strength of our model rather than a limitation. This persistent world-centric memory is precisely what allows FloWM to maintain a coherent representation of the environment under self motion and occlusion, in contrast to methods that do not maintain a persistent latent map and thus frequently hallucinate objects due to their strong generative priors. The intuition for this difference is visualized in our new conceptual figure ([link](https://github.com/AnonFloWM/Flow-Equivariant-World-Modeling/blob/master/new_intuition_figure.png)). The environments we evaluate on are already dynamic and evolving: specifically, the 3D blockworld environment has objects that constantly move, the objects bounce off the walls, and the agent's viewpoint constantly changes under its own self motion. FloWM's stable long horizon performance in these settings is evidence that it can represent and predict the dynamics of changing environments, not just static maps.
>
> That being said, we appreciate the reviewer's comment about expanding environments (e.g. open world exploration), which is a setting that we have not explicitly targeted in our experiments. In our current implementation, the latent memory is recentered as the agent moves and has a fixed spatial extent. However, in principle the map can be extended to represent unbounded maps and worlds. For example, an extension could maintain a coarse-detail global map along with fine-detail local maps, while still applying the same transformations to the current region of the map, analogous in spirit to how hippocampal place cell ensembles allocate distinct maps across environments and scales in biological systems [1, 2, 3]. We see the inclusion of a persistent latent map as a crucial prerequisite for representing disparate parts of such open worlds in a consistent coordinate frame, and we view extending FloWM with multi-scale and dynamically growing maps as an exciting direction for future work.
>
> - [1] The hippocampus as a predictive map: https://www.nature.com/articles/nn.4650
> - [2] Place cells, grid cells, and memory: https://pmc.ncbi.nlm.nih.gov/articles/PMC4315928/
> - [3] Place cells in the hippocampus: Eleven maps for eleven rooms: https://www.pnas.org/doi/10.1073/pnas.1421056111

---

> > ### Author Response · Authors · 2025-11-22
> > **Official Response to Reviewer wAFH (Part 2)**
> >
> > ## Embodiment and environment interaction (W3, W4, Q3)
> >
> > In this work, we broadly use the term embodiment to mean agents that exist within an environment via a body (real or simulated) that receive observations through sensors (e.g. egocentric camera inputs) and take actions that causally affect the environment's state. In our eyes, this does not constrain the consideration of embodiment to only physical robotics tasks, but also includes simulated agent environments, especially those where the viewpoint would emulate that of a real physical environment. We will clarify this in the paper by specifically defining what we mean by embodiment.
> >
> > We believe our environmental setups satisfy this notion: an embodied agent generates the data by moving around and observing only a local window, and the task we consider is to predict future states of the world that this agent would observe after taking actions. We wanted to deliberately isolate partial observability, long horizon memory, and out of view dynamics in the prediction task in this work, and the way that the egocentric observation predictions change given the actions and previous observations is a way of interacting with the environment through navigation.
> >
> > We appreciate the reviewer highlighting the need for studying direct action interactions in future work, such as environments where objects are manipulated directly by the agent. However, we believe that this work serves as a first step towards systems that can consistently model the partially observable dynamics of the physical world, and represents a more promising starting substrate compared to the unstructured baselines we consider. In the paper, we will modify our wording to emphasize that our current focus is on predicting predictable dynamics where the primary agent action is movement, and that FloWM is not a complete robotics system; rather, it is a structured world model whose capabilities in environments with richer interactions (such as robotics datasets) could be studied further by integrating with planning frameworks such as MPC.
> >
> > ## Scalability and environment complexity (Q4)
> >
> > To address concerns about the complexity of our current datasets, we have introduced a new textured variant of the blockworld dataset with new object shapes and visual features. We report the validation rollout results in the table below and see that the FloWM still maintains a significant advantage over baselines despite the increased environment complexity:
> >
> > ## 3D Blockworld with texture metric performance
> >
> > | Model              | MSE ↓      | PSNR ↑  | SSIM ↑  |
> > |--------------------|------------|---------|---------|
> > | **FloWM (Ours)**   | **0.00129** | **28.88** | **0.9342** |
> > | DFoT               | 0.00888    | 20.52   | 0.8588  |
> > | SSM                | 0.01081    | 19.66   | 0.8532  |
> >
> > Blockworld texture reconstruction performance (lower MSE is better; higher PSNR/SSIM are better).
> >
> > Please see the general response to all reviewers above to specifics of the new dataset generation procedure, and visit our anonymous project website [link](https://flowm-anonymous.github.io/Flow-Equivariant-World-Models/) to see qualitative rollouts. These results demonstrate that in more visually complex environments than the original experiments, FloWM can still learn how to represent the world's dynamics in its spatial memory. We hope this evokes confidence in the framework's utility to the community to take a step toward models that can reason about the dynamics of the world without full information.
> >
> > We have not yet evaluated FloWM on physical robotic platforms or real world driving datasets; our current focus is on controlled simulated environments that isolate partial observability and memory. We view applying FloWM to such datasets, combined with stronger perception modules and control layers, as an important direction for future work.
> >
> > ## Concluding remarks
> >
> > We thank the reviewer again for their positive and constructive feedback. In response to their concerns, we have (i) clarified the motion assumptions and highlighted that FloWM already handles non-constant velocities in Blockworld while outlining extensions to more variable and stochastic dynamics; (ii) argued that the persistent latent map is a strength for dynamic environments and discussed how it can be extended to multi-scale, dynamically growing maps in open-ended worlds; (iii) clarified our use of "embodiment" and emphasized the current scope and focus of this paper on partially observable dynamics and prediction of agent navigation; and (iv) added experiments on a more visually complex textured Blockworld, which demonstrate that FloWM continues to maintain coherent dynamics under partial observability. We believe these additions strengthen the case for FloWM as a promising building block for embodied world modeling in more realistic settings. We also invite further discussion.

---

### Official Review · Reviewer_HW2g · 2025-10-29

**Soundness:** 3
**Presentation:** 1
**Contribution:** 2
**Rating:** 4
**Confidence:** 3

**Summary:**

The paper introduces Flow Equivariant World Models (FloWM) for partially observed environments, unifying self-motion and external object motion as one-parameter flows and enforcing equivariance in a latent map. The model maintains a group‑structured memory with velocity channels that co-move with known actions, yielding a stable representation over long horizons. Two instantiations for 2D and 3D scenarios show improved long‑range prediction and length generalization over diffusion baselines. The approach is data‑efficient but currently relies on discretized velocities and a 3D encoder that is only approximately equivariant.

**Strengths:**

S1. Using flows as a unified representation for self‑motion (agent motion) and object motion is interesting and well motivated.

S2. Experiments show strong long‑horizon consistency under partial observability; the model remains stable well beyond the training horizon.

**Weaknesses:**

W1. The problem setup is hard to follow. Please state clearly what is observed, what is predicted, what is latent, and what the training objective is.

W2. The modeling assumptions may narrow the scope of applicable tasks. My understanding (please correct me if mistaken) is that agent actions and “world” actions must be compatible as Lie‑algebra elements; this seems to require both to live in a compatible rigid‑motion group, which may exclude domains where agent actions are non‑rigid or semantic (e.g., crouching, smiling).

W3. All experiments use textureless data; it is unclear how well the approach handles textured or cluttered scenes such as CLEVR/MOVi‑style data.

**Questions:**

Q1. Could this framework be applied to ProcGen‑style games, and if so what adaptations would be needed?

Q2. For driving video generation tasks akin to Wayve's GAIA‑style datasets, what difficulties do you foresee (e.g., multi‑agent stochasticity, occlusions, fine textures, non‑rigid objects)?

Q3. In Fig. 3, is the top‑down latent map h_t provided by the data or inferred from egocentric images?

Q4. In Fig. 2, observations depict overlapping digits while internal flows are defined per digit. Do you first factorize observations with an object‑centric encoder (e.g., slots), or how are digits moved independently without explicit object segmentation?

Q5. How is stochasticity handled? If the world has inherent randomness (e.g., an object moves left or right with equal probability next frame), where does this uncertainty enter the model, and can the method represent multiple futures during prediction?

---

> ### Author Response · Authors · 2025-11-22
> **Official Response to Reviewer HW2g (Part 1)**
>
> ## Summary
> We sincerely thank the reviewer for their time spent with our paper. We appreciate that they consider our work *interesting and well motivated*. We further thank them for acknowledging the substantial improvement to *long horizon consistency* of world models when incorporating flow equivariance -- this is precisely the aspect of world modeling we are most excited about.
>
> In our response, we will directly address the reviewer's concerns by:
> - Clarifying the problem setup and training objective (W1, Q3, Q4)
> - Addressing the scope of the modeling assumptions and applicability to broader tasks such as ProcGen and driving (W2, Q1, Q2)
> - Providing additional experimental results in more visually complex environments (W3, Q2)
> - Discussing how stochasticity and multiple futures can be incorporated into the framework (Q5).
>
> ## Clarifying the problem setup and training objective (W1, Q3)
> We appreciate the opportunity to clarify our exposition. We have taken advantage of this to substantially streamline the text and introduce a new intuition figure available [here](https://github.com/AnonFloWM/Flow-Equivariant-World-Modeling/blob/master/new_intuition_figure.png) to make the overall setup and the role of the latent map more explicit. We acknowledge that due to the fact that we have introduced a general framework for flow equivariant world modeling, and then introduced multiple different instantiations, it can become difficult to disentangle precisely. We include a clarified summary of the experiments and models below for the reviewer's benefit, and will additionally include this in the manuscript.
>
> ### High level overview
>
> All of our experiments follow the same partially observed world modeling setup depicted in Figure 1. Here we provide a high level overview. At each timestep $t$:
> - The agent observes an egocentric RGB frame $f_t$ that covers only a local field of view of the underlying world (a windowed crop in MNIST World, a first person view in Blockworld), together with its own action $a_t$ between timesteps.
> - The model maintains a latent world map $h_t$ (the recurrent hidden state).
> - Given $(f_t, a_t, h_t)$, the model updates $h_t$ and predicts the next observation $\hat f_{t+1}$.
> - We train the model to minimize a next-frame reconstruction loss $\sum_t L(\hat f_{t+1}, f_{t+1})$ (MSE)
> - At evaluation time we roll out autoregressively by feeding predictions back as inputs during the prediction phase
>
> ### Detailed walkthrough
> In the following paragraphs, we provide a more detailed walkthrough of our model. The agent receives only two observations as input at each step $t$: 1) a windowed view of the full world, $f_t$, and 2) its own action between timesteps, $a_t$. On MNIST World, the windowed view of the world is just a fraction of the full 2D world, highlighted in blue in Figure 2; on Blockworld, this observation is the point of view of the agent in the 3D world denoted $f_t$ in Figure 3. The agent receives this windowed view for some number of timesteps (which we denote the "Observation Phase") after which these observations are no longer provided, and the task of the agent is to predict what the future observations should be, given only access to the actions between timesteps. The agent is trained to predict ("roll out") some number of timesteps into the future, and the error is the MSE between these predictions and the true observations. Notably, in order to perform these roll outs, we feed the predicted output observations of the model back in as input for the next timestep ("autoregressively") during this "Prediction Phase".
>
> The hidden state of the model is the primary 'latent' during this procedure. On MNIST, this is simply the hidden state of a convolutional recurrent neural network (visualized as the three vertically stacked feature maps in Figure 2, for the three velocities) -- the network is effectively learning to integrate its limited windowed view of the world into this latent 'world map' over many timesteps. On Block World, the 'latent' is a set of spatially organized tokens (denoted $h_t$ in Figure 3). This is passed between timesteps where the ViT Encoder $E_{\theta}$ acts to effectively update the hidden state with the new observations. We emphasize that this hidden state *is not supervised to match a top-down map*, and *the top down map is never provided to the model as input*. Instead the model has learned to represent the world as something approximating a top down map in order to leverage the flow equivariant recurrent dynamics. We display a top down map in Figure 3 for visualization purposes only so the viewer can understand the environment structure.

---

> ### Author Response · Authors · 2025-11-22
> **Official Response to Reviewer HW2g (Part 2)**
>
> ## Scope of the modeling assumptions and applicability to broader tasks (W2, Q1, Q2)
> The reviewer brings up a valid concern that the current formulation of the Flow Equivariant World Modeling Framework appears to imply that the agents actions and 'world actions' both exist in a compatible Lie Algebra. As we will elaborate below, this is not a necessary condition of the overall theory, but rather a simplification of the framework for the setting where actions of the agent and `world actions' are both rigid motions. We will further describe how, since our model demonstrably works for flows in a learned latent space, the integration of semantic flows becomes much more natural.
>
> Succinctly, under the assumption that the agent has knowledge of its own actions between observations, it is sufficient to apply the self-motion flow separately from the 'internal' flow dimensions. This is because we do not need to 'build in' equivariance to all self-motion flows simultaneously (with loosely termed 'velocity channels'), instead we can simply use the knowledge of the current self-motion to achieve equivariance to that specific flow at each timestep. In other words, as written in Section 3.1, the generalized self-motion flow equivariant recurrence relation may be written as:
>
> $h\_{t+1}(\nu)=T\_{a_t}^{-1}{}\psi\_1(\nu){}\cdot{}\mathrm{U}\_{\theta}\bigl[ h\_{t}(\nu);{}E\_{\theta}\left\[f_t,h_t\right\](\nu)\bigr].$
>
> We see in this setting that the representation of the action $T_{a_t}^{-1}$ can be any linear group representation, and is distinct from the action of the 'internal flows' $\psi_1(\nu)$. It is only in the case that both are translation that we use the combined operator $\psi_1(\nu - a_t)$ for simplicity.
>
> In the case of non-rigid transformations as the reviewer has suggested, these would most likely not have a direct impact on something like visual stimuli, but rather are more likely to impact the agent's observations in a latent space. For example, smiling will not change the direct appearance of the world, but it may change your interpretation of it, and it will likely change how others interact with you. All of these are features of a putative latent space where the representation $T_{a_t}^{-1}$ could serve to transform appropriately. As we have demonstrated in the 3D Block World experiments, our Flow Equivariance framework does indeed work when applied in a learned latent space. Specifically, our model is not supervised to learn a top-down latent map, rather we assert some flow representations in the latent space corresponding to agent actions,  and we observe that the encoder learns an appropriate latent space in order to make use of these flow equivariant dimensions. We therefore suggest there is strong evidence to believe that the same mechanism would be likely to work for other semantic actions with a corresponding appropriate representation in the latent space (e.g. smiling vs. frowning could be represented by a cyclic group of order 2).
>
> In 2D game environments like ProcGen, FloWM can use the same ingredients as in MNIST World. The agent’s view is a limited window of the entire world, and actions shift the agent's view of that world (moving left and right). The velocity channels can naturally track moving entities such as enemies or projectiles, including when they go out of view and reappear later. Further, FloWM could be used for planning with frameworks like MPC in such environments. In driving-style datasets, the latent map plays the role of a world-centric memory of the road scene, with self-motion and the motion of other agents mirroring how FloWM can represent motion of the agent and other blocks in the existing Blockworld experiments. We see both of these environments as interesting future testbeds that fit within the FloWM framework well, and thank the reviewer for mentioning them.

---

> ### Author Response · Authors · 2025-11-22
> **Official Response to Reviewer HW2g (Part 3)**
>
> ## Textures, clutter, and more complex appearance (W3, Q2)
> We sincerely thank the reviewer for highlighting our experimental oversight which could reduce reader confidence in our results. In response, we have added significant variation in shape and texture to our 3D experiments, including changes in block geometry and randomized textures for both objects and environment. In this setting, the encoder must solve a harder observation-to-abstract-world-map mapping problem. We find that FloWM still maintains coherent long horizon rollouts, while DFoT and DFoT-SSM still lose track of out-of-view objects and hallucinate them upon reentry, mirroring the failure modes on the simpler setting. In the table below we report the validation rollout metrics for the three models on this dataset.
>
> ### 3D Blockworld with texture metric performance
> | Model                | MSE ↓     | PSNR ↑  | SSIM ↑  |
> |----------------------|-----------|---------|---------|
> | **FloWM (Ours)**     | **0.00129** | **28.88** | **0.9342** |
> | DFoT                 | 0.00888   | 20.52   | 0.8588  |
> | SSM                  | 0.01081   | 19.66   | 0.8532  |
>
> We provide more details on the dataset generation procedure in our general response to all reviewers, and include new qualitative rollout videos on our project page (https://flowm-anonymous.github.io/Flow-Equivariant-World-Models/).
>
> ## How are objects moved separately? (Q4)
> The reviewer raises the question of how multiple overlapping digits are handled by the model, despite the fact that the flows are defined to operate on the entire hidden state. The reviewer asks if we use a slot-attention-like decomposition or some segmentation to first separate objects.
>
> We clarify that we do not perform any explicit segmentation or slot-like-attention, instead the separation of entities is automatic and implicit in the model, exemplifying the elegance of the approach. In Figure 2, we are visualizing a hypothetical hidden state for a trained 2D FloWM while it is processing a sequence for demonstration purposes.
>
> In detail, due to the 'trivial lift' of the input, the same input is added to all flow channels at each timestep. The resulting 'segmentation' occurs because of constructive interference between a moving input and the corresponding moving hidden state. Informally, the appropriately moving hidden state channel is 'seeing' the input as if it were stationary. In this way, each 'velocity channel' picks up the objects moving with the corresponding velocity without any explicit segmentation or velocity estimation. To decode from this hidden state, we then max-pool over the velocity channels for each pixel independently, giving us the instantaneous observation.
>
> This 'constructive interference' mechanism is identical to the original Flow Equivariant RNN (FERNN) work of Keller (2025), which additionally demonstrates the FERNN is able to model sequences with multiple digits moving simultaneously in different directions. We encourage the reviewer to review Figure 9 in the appendix of that work (https://openreview.net/pdf?id=N1KPOlcN6P) which graphically illustrates this phenomena. There are additional visualizations on the author's blog post associated with the original paper:
> https://kempnerinstitute.harvard.edu/research/deeper-learning/flow-equivariant-recurrent-neural-networks/.

---

> > ### Author Response · Authors · 2025-11-22
> > **Official Response to Reviewer HW2g (Part 4)**
> >
> > ## Stochasticity and uncertainty (Q5)
> >
> > We thank the reviewer for raising this important point. In the current experiments, the environment dynamics are indeed deterministic given the action sequence, and our focus in this work is on a first step: enforcing consistent generation of future dynamics in deterministic settings. In our existing experiments, the context frames contain sufficient information about the objects and their dynamics in order to predict the remainder of the target frames well. Considering the poor results of our strong diffusion baselines, predicting dynamics in environments with deterministic action sequences, which can be viewed as a subset of all possible action sequences, is already a major challenge.
> >
> > We thus see FloWM, which is successful in these deterministic regimes, as a step forward to eventually being able to predict stochastic dynamics and multiple futures. For instance, using a probabilistic generative model within the flow equivariance framework would allow for distributions over trajectories rather than point estimates, providing uncertainty estimates over trajectories which could grow over time. We will add a section in the discussion section calling out this limitation and potential ways forward to deal with stochastic dynamics.
> >
> > ## Concluding remarks
> >
> > In summary, we have clarified the problem setup and training objective (what is observed, what is latent, what is predicted) and explicitly stated that the top-down map is a learned latent representation inferred from egocentric images, not provided by the simulator. We have also refined our explanation of how overlapping objects are handled without explicit segmentation via constructive interference in the velocity channels.
> >
> > We have further clarified the scope of our modeling assumptions, emphasizing that self-motion and internal flows need not share the same group representation and that non-rigid or semantic actions can be modeled as group actions in a latent space. We have added experiments on a more visually complex Blockworld with randomized textures and shapes, demonstrating that FloWM’s advantages persist beyond simple textureless settings, and we have discussed how the framework can be extended to more realistic domains such as ProcGen-style games and driving datasets. Finally, we have outlined how stochasticity and multiple futures can be incorporated into the flow-equivariant framework, and have added a brief discussion of this point to the manuscript’s limitations. We thank the reviewer again for their constructive comments, which have helped improve both the clarity and positioning of the work. We also invite further discussion.

---

> > > ### Comment · Reviewer_HW2g · 2025-11-27
> > >
> > > Thank you for responding to the comments and questions.
> > >
> > > I feel my concern regarding the problem setting is now clarified.
> > >
> > > The new results (videos and accuracies on the textured data) are nice, but the new environment is still similar to the texture-less version (same motions, vivid colors, the object shapes are clearly separated from the background). Also, the prediction errors on the textured dataset are improved from the original one. This sounds strange because I think modeling the textured data is more challenging.
> > >
> > > The authors' comment regarding ProcGen and autonomous driving suggests that there are no critical blockers for applying the proposed method to these more challenging tasks. Then I'd like to see the results to improve confidence in this study.
> > >
> > > Overall, I'm still skeptical of the applicability of the proposed method, and I maintain the original score.

---

> > > > ### Author Response · Authors · 2025-12-03
> > > > **Final response to Reviewer HW2g**
> > > >
> > > > We would like to thank reviewer HW2g for taking the time to respond to our comments, and for the positive reply indicating that their primary concerns about the problem setting are clarified.
> > > >
> > > > We would like to clarify that the image statistics of the textured dataset are different, so the resulting magnitude of the score is not significant. We have now pointed this out clearly in the results we included in our updated manuscript.
> > > >
> > > > While we are excited about the applicability of this method to more complex datasets such as ProcGen and autonomous driving, we leave this to future work for now. The updated manuscript now a more thorough exposition in the limitations section of how the scaling and modeling challenges may be addressed in the future.

---

### Official Review · Reviewer_4jyP · 2025-10-31

**Soundness:** 4
**Presentation:** 3
**Contribution:** 3
**Rating:** 6
**Confidence:** 2

**Summary:**

Summary of Contributions:
This paper introduces a new world model architecture, Flow Equivariant World Models (FloWM), designed for partially observable environments where both the agent and other objects are moving. The core idea is to unify these two types of motion using the principle of flow equivariance. The model maintains a recurrent latent map of the world. This map is explicitly transformed (e.g., shifted or rotated) in the opposite direction of the agent's actions, which provides a stable representation even as the agent moves. In parallel, the model uses a set of "velocity channels" to predict the movement of external objects in the environment, allowing it to track them even when they are outside the agent's limited field of view. The authors show this approach works in both 2D (moving MNIST) and 3D (moving blocks) toy environments.

**Strengths:**

1. The central idea of the paper is very intuitive and compelling. Instead of forcing a large model to re-learn world physics from scratch at every step, the architecture builds in a strong inductive bias for motion. Using an explicit transformation to handle the agent's own movement ("self-motion equivariance") is a clever way to maintain a stable world representation.

2. The experimental results on the provided benchmarks are very strong. The proposed FloWM method significantly outperforms modern baselines like a diffusion-forcing transformer (DFoT) and a state-space model (DFoT-SSM), especially on long-term rollouts (as seen in Figure 4). The baselines clearly fail to track objects that leave the agent's view, while FloWM succeeds.

3. The ablation studies are thorough and effectively demonstrate the value of each component. The model without self-motion equivariance (no SME) fails completely, proving this is the most critical contribution for this task. The model without velocity channels (no VC) also struggles with the dynamic objects, which justifies the full architecture.

4. The abstract and introduction sections are well-written and enjoyable to read. They motivate well injecting the motion-based inductive bias into world models.

**Weaknesses:**

1. The main weakness is the significant and unaddressed question of scalability. All experiments are conducted on simple "toy" datasets (MNIST digits, simple colored blocks). In these settings, the objects are visually simple and come from a small, finite set.
  It is not clear how this method would perform in a real-world scenario, such as autonomous driving or robotics, where the visual space is far more complex and objects are not easily categorized.

2. The 3D Block World experiment requires a ViT encoder to learn a complex projection from a 3D first-person view to a 2D top-down map. The authors note this is not analytically equivariant and must be learned. This projection problem seems like it would become exponentially harder with complex, realistic image data, and this bottleneck could potentially negate the benefits of the equivariant recurrent map.

**Questions:**

1. Could the authors comment on the scalability of this approach? The leap from simple, finite-set objects (digits, blocks) to the visual complexity of real-world scenes (e.g., in driving or robotics datasets) seems very large. What are the primary challenges anticipated?

2. Related to scalability: The 3D-to-2D encoder in the 3D experiment must learn the projection, which is a key potential bottleneck. How well does this projection need to be learned for the flow equivariance to be effective? Have the authors attempted any preliminary experiments on more complex data (e.g., a common dataset like ViZDoom or a driving dataset) to see if this encoder can be trained effectively in a more realistic setting?

3. The current model uses a discrete set of velocity channels (e.g., -2, -1, 0, 1, 2). How would this framework be extended to handle a continuous range of object velocities, as would be common in the real world?

---

> ### Author Response · Authors · 2025-11-22
> **Official Response to Reviewer 4jyP (Part 1)**
>
> ## Summary
>
> We greatly appreciate the reviewer's careful reading of our work and their encouraging assessment. We are particularly glad that they found our framework intuitive and compelling as a step toward accurately and consistently representing dynamics in partially observed worlds, especially over long horizons. Below, we address the reviewer's primary concerns about scalability and environment complexity. Specifically:
>
> - We introduce a new experiment and results on a more visually complex Blockworld (with additional objects and randomized textures/backgrounds) and use it to discuss how FloWM could extend toward more realistic settings.
>
> - We clarify why we view the learned observation-to-world encoder as a strength rather than a bottleneck: it learns an approximately equivariant projection encouraged by the structure of the latent map. We link this to our new textured Blockworld results and discuss how such encoders could scale to more realistic environments.
>
> - We discuss how the current discretized velocity channels can handle continuous velocity ranges in practice and how the framework can be extended toward more continuous flow parameterizations.
>
> ## Scaling visual complexity for partially observable world modeling
>
> We agree that scaling to datasets with more realistic visual and compositional complexity is a key question, and we appreciate the reviewer raising this point explicitly. Our view is that FloWM is primarily a structural contribution: as long as 2D image inputs can be mapped into a consistent latent world coordinate system, the framework can retain its ability to reason about the underlying physical environment.
>
> To better probe this, we have added a new dataset with greater visual complexity, more objects, and randomized textures and backgrounds. In this setting, the encoder must solve a more challenging appearance-to-map projection, moving beyond the simple non-textured original Blockworld experiment. We find that FloWM still successfully learns the observation-to-world projection and retains stable long horizon rollouts, while the baselines DFoT and DFoT-SSM fail in similar ways as before, hallucinating objects without a persistent latent representation of the world. This suggests that the benefits of the flow-equivariant recurrent map are not limited to overly simple or textureless visual regimes. Please see the official comment above for more details about the dataset. Additional quantitative and qualitative results are available on our anonymous website, [link](https://flowm-anonymous.github.io/Flow-Equivariant-World-Models/).
>
> However, we acknowledge that there are many challenges to address before applying FloWM directly to more realistic datasets, such as in-the-wild autonomous driving. We anticipate all the following as reasonable concrete steps for further scaling: (i) stronger visual backbones (e.g. utilizing pretrained ViT models), (ii) more powerful generative decoders (e.g. diffusion based), and (iii) potentially more flexible latent map parameterizations (e.g. point-vector based maps). We see the unified external velocity channel prediction framework and self-motion latent map shift as particularly natural for autonomous driving, where an agent must predict both its own motion and the motion of other drivers and road objects, even when they are temporarily occluded. As for primary challenges, we anticipate at least 3: (i) higher visual complexity and clutter, which calls for stronger visual backbones as mentioned above, (ii) variable / semantic dynamics for other cars and pedestrians (e.g. changing direction while crossing the street), and (iii) richer stochastic representations of other agents. We discuss the extension to continuous ranges of velocities below. We note that the diffusion and SSM baselines we consider, which are closely related to current state-of-the-art architectures, are unable to handle long horizon partial observability even on our simple benchmarks, underscoring the need for structural advances such as FloWM. Overall, we are excited to see future work that scales up the insights from our framework to more complex datasets and larger models.

---

> ### Author Response · Authors · 2025-11-22
> **Official Response to Reviewer 4jyP (Part 2)**
>
> ## Complexity of learning the observation-to-world-map projection
>
> We agree with the reviewer that the lack of a ground truth observation-to-map equivariant encoder is a potential bottleneck for adaptation to more realistic datasets. However, our results indicate that FloWM can learn an effective projection into the latent map, with the structure of the flow equivariant recurrence acting as a regularizer. In our new, more visually and combinatorially complex experiment, FloWM again learns an approximately equivariant encoder, providing evidence that the framework extends to increased visual complexity given sufficient observations of scene transformations.
>
> We also deliberately avoid using ground-truth depth or camera poses for geometric unprojection, both to keep the comparison with diffusion baselines fair (all methods receive only RGB) and to make the method applicable to scenarios without depth sensors. In environments where depth maps and camera poses are available, explicitly unprojecting observations into a 3D representation could provide a strong prior that simplifies learning an accurate encoder. We agree that further empirical study of such encoders on more realistic datasets is an important direction for alleviating this concern.
>
> ## Extending to a continuous range of velocities
>
> We thank the reviewer for this question as it is central to the real world applicability of the model. In the current theory, as the reviewer has correctly identified, the model is only formally equivariant with respect to the discrete set of velocity channels ($V$) that are built into the model. This is nearly identical to how Group Equivariant Convolutional Networks (Cohen & Welling, 2016) discretize the group in order to lift the feature space to the group. In that work, and following work (Weiler & Cesa, 2019; Cesa et al., 2022), it is demonstrated that even with discretization (for example only including 4 or 8 possible rotation angles), the models still exhibit significant performance benefits when evaluated on arbitrarily rotated data (sampled from the full $2\pi$ ). This property has additionally been verified for Flow Equivariant RNNs as well in Figure 6 of the original paper ([link](https://openreview.net/pdf?id=N1KPOlcN6P)), where the model is demonstrated to effectively interpolate between velocities channels when needed. Therefore, while we certainly agree that finding an analog to `steerable' equivariance for continuous velocities would be highly beneficial and very exciting, the current approach is known to already yield significant performance benefits in such practical continuous velocity settings.
>
> ## Concluding remarks
>
> We thank the reviewer again for their close reading and for highlighting the important questions of scalability and visual complexity in our work. In response, we have added experiments on a more visually complex Blockworld with randomized textures and more objects, clarified the role of the learned 3D-to-2D encoder, and discussed how the current discretized velocity channels already can handle continuous velocities in a manner analogous to group-equivariant CNNs, with clear paths toward more continuous formulations.
>
> We view FloWM as a structural contribution: a flow-equivariant recurrent world map that maintains coherent memory under self-motion and out-of-view dynamics. Our new experiments suggest that this structure extends beyond simple visual regimes, and we believe it provides a natural foundation for future work that combines stronger visual backbones and more expressive velocity parameterizations in realistic domains such as driving and robotics. We also welcome any further discussion.

---

### Official Review · Reviewer_3j6v · 2025-11-01

**Soundness:** 2
**Presentation:** 2
**Contribution:** 2
**Rating:** 2
**Confidence:** 4

**Summary:**

The paper proposes Flow Equivariant World Models (FloWM) that treat both self-motion and external object motion as Lie-group flows. It derives a generalized flow-equivariant recurrence where dedicated tokens encode self motion and internal flows track external motion. Benchmarks are a synthetic 2D MNIST World and a 3D Dynamic Block-World evaluated up to multiple rollout steps. The authors show lower error than diffusion baselines in terms of generative quality metrics (MSE and SSIM) and better long-horizon generation.

**Strengths:**

1. The method provides a clear formalization of flow equivariance for both self and external motion in a theoretically grounded world modeling framework. Methods that can model latent actions/causes/dynamics that are external to agent's own action are welcome and needed in the era of world models.

2. The method targets partial observability with out-of-view dynamics, which is a gap in video models that can suffer from limited context windows.

**Weaknesses:**

1. Beyond MNIST-World and the 3D Block benchmarks which are designed by authors, there is no evaluation on external benchmarks.

2. Diffusion baselines sometimes underperform even vs trivial ablations and all-black baseline (table 1), which highlights my point above that the benchmarks might favor FloWM’s design. Table 1 and 2 shows diffusion is worse than “no SME, no VC” (i.e. a vanilla ConvRNN) on most metrics with similar pattern in Table 2, which questions the fairness of evaluation protocol and effectiveness of VC and SME.

3. The evaluation metrics are based on generative quality only. Since the method is in the domain of equivariant representation learning, such metrics are not a good proxy for decoding actions directly. For example, one evaluation can decode actions from latents corresponding to velocity channels.

4. Baselines are limited to diffusion models. Other world modeling paradigms (e.g. JEPAs [1] or masked video models [2]) are not compared with.


[1] V-jepa 2: Self-supervised video models enable understanding, prediction and planning.

[2] VideoMAE: Masked Autoencoders are Data-Efficient Learners for Self-Supervised Video Pre-Training

**Questions:**

1. What is your explanation for the fact that a simple ConvRNN (“no SME, no VC”) sometimes outperforms baselines in Tables 1 and 2?

2. Can you report compute comparisons (FLOPs and wall-clock time) to baselines?

---

> ### Author Response · Authors · 2025-11-22
> **Official Response to Reviewer 3j6v (Part 1)**
>
> ## Summary
>
> We sincerely thank reviewer 3j6v for their time spent with our manuscript. We appreciate that they acknowledge the importance of our work in providing a theoretically grounded solution to two significant gaps in the current world modeling literature: external world motion, and partially observed dynamics.
>
> In our response below, we reply to weaknesses and questions that the reviewer identified, including new experiments which directly address their concerns. Concretely:
>
> - We explain why, on certain subsets, a simple ConvRNN and even the all-black baseline can achieve better MSE than more expressive models, and show that this is a consequence of finite-context limitations and metric sensitivity, not a flaw in baseline capacity or training.
>
> - We discuss the reviewer’s concern about additional baselines and use a new intuition figure to argue that our existing diffusion-based baselines are representative of current finite-context video generative models that lack persistent memory.
>
> - We add FLOP and runtime comparisons indicating that, while FloWM is modestly more expensive to train than our baselines, it delivers much more stable long horizon predictions under partial observability.
>
> ## Poor Baseline Performance
>
> We understand the reviewer’s concern regarding the apparent poor performance of some baselines. However, as we detail below, this behavior is not due to insufficient model capacity or flawed training, but reflects a structural limitation of finite-context, sliding-window world models on our partial observability benchmarks. Specifically, finite-context models cannot maintain a persistent memory of objects once they leave the current context window, and thus eventually revert to hallucinating plausible content based on their strong generative priors. By simply predicting the dominant background (as in the all-black baseline), a trivial model can sometimes obtain better metric scores (e.g. MSE) than the more realistic predictions, despite being useless as a world model. We therefore include additional visualizations comparing ground truth frames, all-black predictions, and model predictions, together with their differences and MSEs. See [link](https://github.com/AnonFloWM/Flow-Equivariant-World-Modeling/blob/master/mse_fig.png) for intuition on how hallucinated predictions can lead to a worse MSE score, and see [link](https://github.com/AnonFloWM/Flow-Equivariant-World-Modeling/blob/master/mse_fig_2.png) for intuition on how all-black can sometimes produce misleadingly good metric scores on this benchmark. Our motivation for including all-black was to attempt to normalize the prediction metrics for rollouts of different lengths.
>
> We would like to emphasize that the metric score must be very good to indicate it is a good predictive model, and so the comparison between two poor scores, such as the ones highlighted by the reviewer between diffusion and a vanilla ConvRNN, does not hold significant meaning. Rather, they are both poor scores, and so both models can be seen as failing the task. We strongly encourage the reviewer to view the comparative qualitative videos of the model rollouts, visible on our project page to see visual examples of these phenomena in practice. (Videos: [link](https://flowm-anonymous.github.io/Flow-Equivariant-World-Models/), Figure: [link](https://raw.githubusercontent.com/AnonFloWM/Flow-Equivariant-World-Modeling/refs/heads/master/blockworld_dynamic_rollouts.png)).

---

> > ### Author Response · Authors · 2025-11-22
> > **Official Response to Reviewer 3j6v (Part 2)**
> >
> > ## Additional Baselines
> >
> > We agree with the reviewer that additional baselines are always valuable; however, as outlined in the above discussion and outlined in our new intuition figure, available here ([link](https://github.com/AnonFloWM/Flow-Equivariant-World-Modeling/blob/master/new_intuition_figure.png)), the poor performance we observe stems from the lack of a persistent memory rather than from the particular choice of generative model. While we have not run all possible baselines, we note that many strong video models (including VideoMAE, mentioned by the reviewer) are designed around fixed-length context windows and do not maintain a structured world memory, so we anticipate that they would exhibit qualitatively similar failure modes without further modification. So while it is true that we limit our comparison to diffusion-based models, we argue that the focus should be on the ability for the models to represent latent dynamics, instead of the way the decoder generates.
> >
> > We appreciate the suggestion to compare against latent world modeling methods such as V-JEPA 2. However, V-JEPA 2 is trained with a self-supervised, non-generative objective and is not directly designed for multi-step rollout generation, which is the focus of our evaluation. We view the FloWM framework as complementary to such approaches, and an exciting future direction would be to combine the FloWM and V-JEPA frameworks. In such a setting, the flow equivariant latent map could in principle be used as the representation backbone within a JEPA-style objective, or combined with masked video pretraining. We believe this would be very exciting future work which our proposed theory would facilitate.
> >
> > We thank the reviewer for pointing out these works and will include a detailed discussion of how our work relates to them in the final version of the paper.

---

> > > ### Author Response · Authors · 2025-11-22
> > > **Official Response to Reviewer 3j6v (Part 3)**
> > >
> > > ## Computational Comparison
> > >
> > > We appreciate the reviewer's attention to the scalability and applicability of our method, and agree that it is important to normalize resources as much as possible to ensure that they are on even playing field. Using a recurrent architecture like ours trades away some computational efficiency (especially with modern deep learning hardware), but we can see in our case that we get back significantly improved generation quality. In the tables below, we report the training compute and inference wall clock time of the FloWM and baselines for both training and inference. Benchmarking is done on 1 NVIDIA H100 GPU. Total training compute (EFLOPs) are calculated by measuring the forward / backward GFLOPs measured for batch size = 1, then the estimates are scaled by the actual batch size and number of training steps. We observe that FloWM requires roughly 1.7–2.6× more training FLOPs than DFoT and DFoT-SSM to reach convergence, but remains within the same order of magnitude while delivering substantially more stable long horizon performance.
> > >
> > > Inference wall clock time is reported for one frame prediction with batch size = 1. These per-step runtimes are also of the same order of magnitude across models, as seen in the table below. FloWM is not parallelizable over sequence length because it maintains a recurrent world state, so inference scales linearly with the number of predicted frames. We view this linear dependence as a tradeoff for preserving a coherent latent map over long horizons, possibly a necessary tradeoff to accurately represent hidden dynamics.
> > >
> > > Our current code implementation is likely not fully optimized, and we expect that there is substantial room for speedup. We briefly highlight three directions that could further reduce the computational cost without changing the core architecture: (1) Decoupling the encoder $E_\theta$ from the hidden state $h_t$ would allow parallel processing of input frames before writing to the map, enabling greater input-level parallelism; (2) The recurrent update mechanism could be made linear and associative, which when combined with (1) would allow for application of the parallel scan algorithm leveraged by modern state space models (SSMs) enabling parallelization over sequence length [1, 2]; (3) the latent map update is currently a dense gated update over the full estimated field of view, whereas the true information change per timestep is likely sparse. Introducing sparse or multiscale updates, for example by maintaining a coarse-to-fine feature hierarchy and updating only affected regions, could significantly reduce per-step computation while preserving the benefits of a persistent world memory (inspiration could be taken from existing hierarchical mapping methods [3, 4]).
> > >
> > > - [1] Parallelizing Linear Recurrent Neural Nets Over Sequence Length. Martin and Cundy, ICLR 2018. ([link](https://arxiv.org/abs/1709.04057))
> > >
> > > - [2] Simplified State Space Layers for Sequence Modeling. Smith et al. ICLR 2023. ([link](https://arxiv.org/abs/2208.04933))
> > >
> > > - [3] NICE-SLAM. Zhu et al. CVPR 2022. ([link](https://arxiv.org/abs/2112.12130))
> > >
> > > - [4] SHINE-Mapping: Zhong et al. ICRA 2023. ([link](https://arxiv.org/abs/2210.02299))
> > >
> > > ## Training benchmarking
> > >
> > > | **Model**       | **Batch size** | **Steps** | **Total compute (EFLOPs)** |
> > > |----------------|----------------|-----------|----------------------------|
> > > | FloWM (Ours)   | 16             | 150k      | 27.5                       |
> > > | DFoT           | 32             | 300k      | 15.7                       |
> > > | DFoT-SSM       | 32             | 300k      | 10.5                       |
> > >
> > > ## Inference benchmarking
> > >
> > >
> > > | **Model**       | **Forward Time (ms)** | **Backward Time (ms)** |
> > > |----------------|------------------------|-------------------------|
> > > | FloWM (Ours)   | $104.47 \pm 101.05$    | $72.35 \pm 2.37$        |
> > > | DFoT           | $90.93 \pm 109.59$     | $106.94 \pm 4.70$       |
> > > | DFoT-SSM       | $47.52 \pm 0.85$       | $89.85 \pm 0.42$        |

---

> ### Author Response · Authors · 2025-11-22
> **Official Response to Reviewer 3j6v (Part 4)**
>
> ## External Benchmarks
>
> The reviewer additionally raises the concern that our method was not evaluated on external benchmarks, raising questions about the generality of our method and results. To address this concern, we first note that our benchmarks were chosen to explicitly test for the conceptual limitation of existing SoTA world models for handling partially observed dynamics, illustrated in our new intuition figure:  ([link](https://github.com/AnonFloWM/Flow-Equivariant-World-Modeling/blob/master/new_intuition_figure.png)). In searching, we found that no existing external benchmarks provided the same level of partially observed dynamics necessary to highlight the limitations of existing models and demonstrate how Flow Equivariance alleviates this limitation. For example the vast majority of benchmarks only have very limited external dynamics, or only dynamics \emph{in the field of view} which do not require a long term modeling of these external systems. Yet, natural environments certainly do display partially observed dynamics, such as those originating from external agents, and thus we believed developing new benchmarks to test this property explicitly in a controlled manner was warranted.
>
> To reach a middle ground which makes our experiments more similar to existing external benchmarks and hopefully further address the reviewer's concern, we have provided new experimental results where we increased the visual complexity of our Blockworld dataset to align it more closely with the commonly used external benchmarks such as VizDoom and DMLab. Specifically, we have added additional shapes, and randomized textures across both moving shapes and the background. In this setting, the encoder is tasked with a significantly harder recognition problem (similar to other benchmarks, but with significantly greater partially observed dynamics); yet, as we see in the table below, the FloWM continues to outperform diffusion baselines on long horizon rollouts, suggesting that our advantages are not tied to overly simple or textureless environments or specific object shapes.
>
> The reviewer can view full rollout videos across multiple differently sampled environments on our website [link](https://flowm-anonymous.github.io/Flow-Equivariant-World-Models/), and see a concise figure comparing rollouts on this new dataset here ([link](https://raw.githubusercontent.com/AnonFloWM/Flow-Equivariant-World-Modeling/refs/heads/master/blockworld_tex_rollouts.png))
>
> ## 3D Blockworld with texture metric performance
>
> | Model            | MSE ↓      | PSNR ↑ | SSIM ↑  |
> |------------------|------------|--------|---------|
> | **FloWM (Ours)** | **0.00129** | **28.88** | **0.9342** |
> | DFoT             | 0.00888    | 20.52  | 0.8588 |
> | SSM              | 0.01081    | 19.66  | 0.8532 |
>
> ## Concluding remarks
>
> In summary, we have clarified why diffusion baselines, simple ConvRNN ablations, and the all-black baseline can sometimes produce surprising results for the metrics we consider, and we have supplemented our quantitative results with new visualizations that provide intuition for this phenomena. We have also provided FLOP and runtime comparisons showing that FloWM’s gains in long horizon stability and accuracy come at a modest increase in compute relative to diffusion baselines, and we have outlined concrete avenues for further efficiency improvements. Lastly, we have provided justification for the current baselines and benchmarks we consider, and have presented additional results on more complex environments to address the reviewer's concerns about the broader applicability of our approach.
>
> We will certainly integrate these discussions and considerations into our updated manuscript, and thus thank the reviewer for helping us to improve our paper. We welcome any further discussion.

---

> ### Comment · Reviewer_3j6v · 2025-11-27
>
> Thank you for your response.
>
> The authors have made their case well and addressed most of my concerns. I increased my score to 4. However, W3 has not been addressed yet. I know the method is generative, but you can perform an evaluation based on the protocol common in equivariant SSL literature [1, 2, 3] for your model and baselines, i.e., feeding the frozen concatenated representations of $x_t$ and $x_{t+1}$ to an MLP to decode the action.
>
> Also, V-JEPA 2 [4] is capable of autoregressive rollout and planning (e.g. their planning experiments). Although a direct comparison is not required right now, the possibility of adopting the framework for action-aware representations in JEPA world models should be mentioned as future work.
>
>
> [1] Self-supervised learning of Split Invariant Equivariant representations
>
> [2] In-Context Symmetries: Self-Supervised Learning through Contextual World Models
>
> [3] seq-JEPA: Autoregressive Predictive Learning of Invariant-Equivariant World Models
>
> [4] V-JEPA 2: Self-Supervised Video Models Enable Understanding, Prediction and Planning

---

> > ### Author Response · Authors · 2025-12-03
> > **Final response to reviewer 3j6v**
> >
> > We greatly appreciate reviewer 3j6v for indicating they will raise their score and for the positive comment that "the authors have made their case well and addressed most of my concerns."
> >
> > Regarding the suggestion to use the SSL protocol from the papers cited (predicting actions from concatenated neighboring hidden states): we believe with our model and experimental setup, this is not an appropriate additional evaluation. In our architecture, the action is represented separately from the velocity channels, which are used to model external object motion. In addition, our FloWM action shift operation already makes the memory framework analytically equivariant to self motion by design, so this protocol does not provide additional insight. We emphasize that the focus of this method is indeed on the accuracy of the generated video, which is the reason why we chose to focus on those as our evaluation criteria.
> >
> > We thank the reviewer for emphasizing the interesting potential synergies involving latent world models (e.g. JEPA) and our framework. We have added a discussion of such methods and future work in our limitations and future work section.

---

### Author Response · Authors · 2025-11-22
**Summary of responses to concerns from first round of reviews**

## Summary

We greatly appreciate the time, effort, and care that all of the parties have spent in reviewing our work.

We appreciate that reviewers have collectively described our work as 'theoretically grounded' (3j6v); 'clever', 'intuitive', and 'compelling' (4jyP); 'interesting and well motivated' (HW2g); and 'elegant' and 'rigorous' (wAFH). Reviewers have additionally praised our work for theoretically unifying self and external motion (3j6v, HW2g, wAFH); addressing a gap in the current world modeling literature (3j6v, wAFH); performing thorough ablation studies (4jyP, wAFH); and ultimately showing strong experimental results (4jyP, HW2g, wAFH).

At a high level, we believe the reviewers' concerns are captured in three broad categories:

- Scalability of our approach to more realistic environments.
- Theoretical limitations of our proposed framework.
- Details of model specifics and evaluation metrics.

To directly address these points, in our responses to the reviewers below, we have:

- Provided new experimental results, evaluating our model on a significantly more complex 3D visual environment with varying shapes and textures; demonstrating consistent strong performance over baselines (details in the following section).
- Expanded on the full scope of the Flow Equivariant World Modeling framework, illustrating that both the theory and practical benefits are in fact broader than the specifics of the instantiations presented in this work.
- Introduced new figures and videos to:
  - Visualize rollouts on the new datasets:
    - Videos: (<https://flowm-anonymous.github.io/Flow-Equivariant-World-Models/>)
    - Figure: (<https://github.com/AnonFloWM/Flow-Equivariant-World-Modeling/blob/master/blockworld_tex_rollouts.png>)
  - Concretely visualize performance on the original Blockworld dataset:
    - Figure: (<https://github.com/AnonFloWM/Flow-Equivariant-World-Modeling/blob/master/blockworld_dynamic_rollouts.png>)
  - Clarify the conceptual limitations of existing SoTA, and how our framework resolves this
    - Figure: (<https://github.com/AnonFloWM/Flow-Equivariant-World-Modeling/blob/master/new_intuition_figure.png>)
  - Provide visual examples to ground the interpretation of quantitative metrics:
    - Figure: (<https://raw.githubusercontent.com/AnonFloWM/Flow-Equivariant-World-Modeling/refs/heads/master/mse_fig.png>)
    - Figure: (<https://raw.githubusercontent.com/AnonFloWM/Flow-Equivariant-World-Modeling/refs/heads/master/mse_fig_2.png>)

We again thank the reviewers for their role in helping us improve our manuscript and welcome further discussion.

---

## New Experimental Results on More Realistic Environment

In order to address concerns about scalability and complexity of the datasets considered in our paper, we introduce Textured Blockworld, a visually complex and more realistic dataset with the same partially observable dynamics properties as the original Blockworld dataset. Specifically, the objects can now be blocks or spheres, and the blocks have one of three textures chosen for their surface. The number of objects spawned stays the same as before, randomly chosen between 6 and 10. In addition to the object texture, the walls and floor are also randomly selected from 3 different textures each. Taken together, these randomization aspects allow for exponentially large combinations of potential room configurations.

Quantitative results are available in the table below for 150-frame rollouts given 50 frames of context, and qualitative rollouts are visible on the website [link here](<https://flowm-anonymous.github.io/Flow-Equivariant-World-Models/>). We find that FloWM still successfully solves the task with significantly improved performance compared with the baselines. Specifically, we see the FloWM again learns the appropriate approximately-equivariant 3D-to-2D projection and retains stable long horizon rollouts, while DFoT and DFoT-SSM continue to lose track of out-of-view objects and hallucinate them when they re-enter, reflecting the same memory limitations observed on the original datasets.

## 3D Blockworld with texture metric performance

| Model            | MSE ↓      | PSNR ↑ | SSIM ↑  |
|------------------|------------|--------|---------|
| **FloWM (Ours)** | **0.00129** | **28.88** | **0.9342** |
| DFoT             | 0.00888    | 20.52  | 0.8588 |
| SSM              | 0.01081    | 19.66  | 0.8532 |

---

### Author Response · Authors · 2025-12-03
**Summary of rebuttal period for AC**

## High level summary

Dear AC,

We thank you for overseeing the review process and for taking the time to read and evaluate our work, especially considering the additional challenges of this year's ICLR review process. Here, we would like to summarize (i) the main contributions of our work, (ii) how the rebuttal period clarified several initial misunderstandings, (iii) why the remaining concerns raised by the reviewers largely reflect reasonable directions for future work rather than fundamental issues with our current results, and (iv) the concrete additions and revisions we have already made to the manuscript.

## Core contribution and significance of FloWM

FloWM introduces a structural advance in video world modeling, learning a recurrent latent memory well suited for representing and predicting partially observed dynamics. FloWM's flow equivariant recurrent world map unifies self motion of an embodied agent with external object motion as Lie group flows in a shared latent memory in order to predict future observation frames with high fidelity.

Critically, this recurrent latent map is 'self-motion equivariant' via latent map shifts that ensure that the world representation remains stable under egocentric viewpoint changes. FloWM is further equivariant to the motion of external objects through the addition of flow-equivariant 'velocity channels', directly modeling external flows and tracking objects even when they leave the field of view, addressing the key gap in current video world models under partial observability.

Across both 2D MNIST World and 3D Block World environments, FloWM consistently delivers substantially improved long horizon rollouts in comparison to strong diffusion transformer and SSM world model baselines. In this work, we focus on the setting of deterministic dynamics and isolate the problem of partial observability, targeting the regime in which current finite context video models fail.

We emphasize that the focus of this paper is to demonstrate how the theory of flow equivariance may leveraged to tackle the problem of partially observed dynamic world modeling, and to highlight that current popular architectures are not well suited, rather than focusing on the introduction of  a full fledged state of the art world modeling architecture at this time. Through the additional experiments we have introduced during the rebuttal period that increase the visual complexity, we hope the separation between structural advances and task difficulty is clearer; suggesting that our theoretical contributions are indeed likely to extend to more realistic scenarios, thereby making a significant contribution to the current state of the world modeling literature. Overall, we believe FloWM represents an exciting and principled step forward toward world models that can accurately represent and predict the dynamics of the world over long horizons under partial observability.

---

> ### Author Response · Authors · 2025-12-03
> **Summary of rebuttal period for AC (cont.)**
>
> ## Key clarifications made during the rebuttal period
>
> The initial round of reviews raised strong and constructive concerns, but also revealed that several key aspects of the setting and framework were not fully clear to all reviewers. We used the rebuttal period to refocus attention on the actual goal of the paper: generative world modeling under partial observability with persistent memory. Additionally, we presented results on a new, more visually complex environment, introduced multiple new figures, and updated the paper with further model and result clarifications to address limitations.
>
> Specifically, to address concerns about scalability and complexity of the datasets in our paper, we introduced Textured 3D Block World, which has randomly selected textures and more visual complexity, while retaining the partially observable dynamics as the focus of the environment interactions. Quantitative results are available in the table below, and qualitative rollouts are visible on the website [link here](<https://flowm-anonymous.github.io/Flow-Equivariant-World-Models/>). FloWM still successfully solves the task despite the additional visual complexity, while the baselines still retain the same failure modes; thereby providing evidence for the separation of our theoretical contributions from the practical technical considerations of state of the art world modeling environments.
>
> ### Textured 3D Block World metric performance
> | Model            | MSE ↓      | PSNR ↑ | SSIM ↑  |
> |------------------|------------|--------|---------|
> | **FloWM (Ours)** | **0.00129** | **28.88** | **0.9342** |
> | DFoT             | 0.00888    | 20.52  | 0.8588 |
> | SSM              | 0.01081    | 19.66  | 0.8532 |
>
> The conceptual clarifications revolved around metric calculation, the reason for choosing our baselines, assumptions made in our model, and more. A more exhaustive list of clarifications that we have already applied to the updated manuscript is provided in the section below. Further, we introduced new rollout figures to make visual discrepancies between model-generated frames and the ground truth more clear (links [here](<https://github.com/AnonFloWM/Flow-Equivariant-World-Modeling/blob/master/blockworld_dynamic_rollouts.png>) and [here](<https://github.com/AnonFloWM/Flow-Equivariant-World-Modeling/blob/master/blockworld_tex_rollouts.png>)), as well as a new conceptual figure comparing our model to the baseline [link here](<https://github.com/AnonFloWM/Flow-Equivariant-World-Modeling/blob/master/new_intuition_figure.png>).
>
> The subsequent interactions with the reviewers show that these clarifications were effective:
>
> - Reviewer 3j6v wrote that "the authors have made their case well and addressed most of my concerns" and indicated they would increase their score.
>
> - Reviewer HW2g explicitly stated that "my concern regarding the problem setting is now clarified."
>
> - Reviewers 4jyP and wAFH have consistently emphasized that the core idea is intuitive, elegant, and theoretically grounded, and that our experiments demonstrate strong long-horizon consistency. Unfortunately they did not have the chance to respond, but we believe their concerns would have been well addressed by our additional results and exposition.
>
>
> ## Scope of remaining concerns
>
> The remaining reservations brought up by the reviewers are now primarily about future scalability (e.g. application to datasets such as ProcGen, autonomous driving, large scale robotics), not about the soundness of the framework or the validity of our current experimental results. We would like to emphasize once again that our core contribution is not to solve a wide range of real world benchmarks at this time, but is instead to present a novel framework to address the problem of partially observable dynamic world modeling.
>
> Specifically, we felt that the additional concern raised by reviewer 3j6v about self supervised learning evaluations would not be an appropriate additional evaluation, as the velocity is represented separately from the action representation. Further detail is provided in our final response to that reviewer. Reviewer HW2g's final concerns about the scale of benchmark scores have also been addressed in a response to their comment.

---

> > ### Author Response · Authors · 2025-12-03
> > **Summary of rebuttal period for AC (cont.)**
> >
> > ## Concrete changes and additions to the paper
> >
> > To respond to the detailed concerns mentioned in the rebuttal process, we have already made substantial updates to the manuscript and experimental suite. Concretely, we:
> >
> > - Introduce details, results, and analysis on a more visually complex 3D Textured Block World dataset
> >
> > - Add a new intuition figure to clarify differences between baselines and our model.
> >
> > - Add new rollout figures to emphasize the visual quality of FloWM’s predictions and the hallucinations exhibited by the baselines.
> >
> > - Add training and inference compute analysis to the appendix.
> >
> > - Improve exposition of the model section to be more specific about the assumptions of the model in the text.
> >
> > - Significantly expanded the limitations and future work section to include many concerns mentioned by the reviewers, including: rigidity of the action parameterization and extensions to semantic actions; clarifications on the deliberate focus of the work on deterministic dynamics and a path forward to modeling stochastic dynamics; limitations of the fixed size map and paths forward for handling open world environments; synergies between FloWM and latent world modeling frameworks such as JEPA to do planning for tasks such as autonomous driving or robotics; and charting realistic efficiency improvements to address scalability concerns.
> >
> >
> > ## Final Summary for the AC
> >
> > In summary, the rebuttal process has, in our view, substantially improved both the clarity and positioning of FloWM. Initial confusion about the problem setting, the nature of the latent map, and the scope of our motion assumptions has been resolved, as acknowledged in reviewers’ followup comments. We have added a more complex dataset, new figures and videos, detailed compute comparisons, and expanded theoretical and related-work discussions, all of which aim to make the case for FloWM as a robust, well-founded approach to partially observed world modeling. The remaining criticisms now center on further scaling to very challenging real-world benchmarks. These are directions that we would like to frame as natural next steps, rather than as requirements for the present contribution.
> >
> > We therefore respectfully submit that FloWM represents a substantive step forward. FloWM offers a principled way to endow world models with structured, flow equivariant memory that remains stable under self-motion and out of view dynamics. FloWM empirically demonstrates the limitations of current finite context generative baselines in precisely this regime. We hope you will weigh this structural contribution, together with the clarified manuscript and new results, favorably in your decision.

---

### Meta-Review · Area_Chair_ApiM · 2026-01-07

**Summary:**

Three of the four reviewers acknowledged the work’s strong theoretical grounding. One reviewer remained skeptical, primarily due to the absence of evaluations on established external benchmarks and concerns about scalability to visually complex or stochastic real-world settings. The authors addressed nearly all technical concerns through extensive new experiments, visualizations, and manuscript revisions. However, based on the opinions of all reviewers, the paper did not meet acceptable standards.

**Reviewer Concerns:**

The authors provided thorough responses to all raised weakness and questions.

**Reviewer Scores:**

Reviewer 3j6v (initial 2) raised their score to 4 after the authors clarified the problem setting, added textured results, and addressed baseline concerns.
Reviewer 4jyP (initial 6) would likely maintain or slightly reinforce their score, as their questions on scalability and velocity discretization were well answered with evidence of interpolation and pathways to continuous extensions.
Reviewer HW2g (initial 4) maintained their score, acknowledging clarification of the setup.
Reviewer wAFH (initial 4) would likely raise or maintain their score.

---

### Decision · Program_Chairs · 2026-01-26

Reject